# Current Progress in Femtosecond Laser Ablation/Ionisation Time-of-Flight Mass Spectrometry

**Marek Tulej \*, Niels F.W. Ligterink, Coenraad de Koning, Valentine Grimaudo** **, Rustam Lukmanov, Peter Keresztes Schmidt, Andreas Riedo and Peter Wurz**

Physics Institute, University Bern, Sidlerstrasse 5, 3012 Bern, Switzerland; niels.ligterink@csh.unibe.ch (N.F.W.L.); coenraad.dekoning@space.unibe.ch (C.d.K.); valentine.riedo@space.unibe.ch (V.G.); rustam.lukmanov@space.unibe.ch (R.L.); peter.keresztes@space.unibe.ch (P.K.S.); andreas.riedo@space.unibe.ch (A.R.); peter.wurz@space.unibe.ch (P.W.)
\* Correspondence: marek.tulej@space.unibe.ch

**Abstract:** The last decade witnessed considerable progress in the development of laser ablation/ionisation time-of-flight mass spectrometry (LI-TOFMS). The improvement of both the laser ablation ion sources employing femtosecond lasers and the method of ion coupling with the mass analyser led to highly sensitive element and isotope measurements, minimisation of matrix effects, and reduction of various fractionation effects. This improvement of instrumental performance can be attributed to the progress in laser technology and accompanying commercialisation of fs-laser systems, as well as the availability of fast electronics and data acquisition systems. Application of femtosecond laser radiation to ablate the sample causes negligible thermal effects, which in turn allows for improved resolution of chemical surface imaging and depth profiling. Following in the footsteps of its predecessor ns-LIMS, fs-LIMS, which employs fs-laser ablation ion sources, has been developed in the last two decades as an important method of chemical analysis and will continue to improve its performance in subsequent decades. This review discusses the background of fs-laser ablation, overviews the most relevant instrumentation and emphasises their performance figures, and summarizes the studies on several applications, including geochemical, semiconductor, and bio-relevant materials. Improving the chemical analysis is expected by the implementation of laser pulse sequences or pulse shaping methods and shorter laser wavelengths providing current progress in mass resolution achieved in fs-LIMS. In parallel, advancing the methods of data analysis has the potential of making this technique very attractive for 3D chemical analysis with micrometre lateral and sub-micrometre vertical resolution.

**Keywords:** chemical composition; elemental imaging; sub-micrometre resolution; laser ablation; time-of-flight; mass resolution; atomic ions

## 1. Introduction

LIMS has a rich history spanning more than 50 years. The initial fast expansion lasted from the middle of the 1970s to the beginning of the 1990s, when it offered the most affordable and fast method of analysing solid samples, including organic and bioorganic compounds [1–3]. Several commercial instruments such as LAMMA-500, -1000, -2000 became available in the 1970s [2,4–8]. Its relative decline in the 1990s was caused by the development of more efficient ionization methods, including electron spray ionisation (ESI), glow discharge (GD), matrix-assisted laser desorption (MALDI), and laser ablation inductively coupled plasma (LA-ICP) ion sources, the last two of which represent descendants of LIMS. These ion sources proved more reliable than the laser ablation ion sources of the time and offered means for high accuracy and precision in isotope and element measurements [9,10]. In recent years, LIMS has experienced a revival as a mass spectrometric method. The development of high irradiance and high fluence laser instrumentation, a

miniature mass analyser for space research application, and a high mass resolution laboratory instrument within about one decade opened a new perspective for this technique. The use of fs-lasers made LIMS a reliable and high precision method for depth profiling and imaging of surfaces of inorganic samples with high boiling point temperatures, in a more compact and less complex setup than comparable LA-ICP-MS and SIMS instruments. The simplicity of LIMS instrumentation allows for the design of very compact analytical instruments suitable for element and isotope analysis of geological and planetary samples, thus offering its use for application on future planetary missions. The high performance of current LIMS instruments for the analysis of solid samples was demonstrated in numerous studies, as well as several selected applications e.g., in material science, geology and bioanalytics. Furthermore, LIMS can be used to study molecules present on the surface by applying a direct laser desorption method [11–14]. Recently, such an instrument was proposed for the measurements of amino acids on Jupiter's moon Europa [12].

The present review discusses the mass spectrometric studies conducted by LIMS instrumentation employing a femtosecond laser ablation ion source and a time-of-flight (TOF) mass separator. The historic evolution of (ns-)LIMS, various instrument designs, and numerous applications can be found in several earlier reviews and books [2,3,15–17]. Additionally, we do not consider other important techniques that use laser-based probes, e.g., Fourier Transform Ion Cyclotron Resonance (FT-ICR), Orbitrap or Resonance Ionisation Mass Spectrometry (RIMS). An overview of these studies can be found in the recent review by Azov et al. [1]. Our focus on studies with fs-laser ablation source is motivated by its high impact on the quantitative performance and superior chemical imaging capabilities of LIMS. The TOF mass analyser offers suitable coupling to the pulsed ion source and multi-elemental measurement capabilities. New high-performance LIMS instruments were designed in the 2000s and implemented fs-laser ablation ion sources in the late 2000s [18] and the beginning of the 2010s [19–21]. They demonstrated their high sensitivity and capabilities to perform high-resolution depth profiling [19,22,23]. Improved mass resolution compared to previously known instruments was achieved in an instrument with an orthogonal ion extraction [23], and later with co-linear extraction [24] source. Advances in the development of fast ion detectors, data acquisition systems, and improvement of the modelling tools helped the development of miniature LIMS systems and high-resolution laboratory LIMS systems [24–26]. Without computer-assisted development tools, the detailed instrument design and optimisation of its operation would be difficult. Without advanced modelling tools, it would also be difficult to arrange an optimal optical system for the ion source and the optical microscope frequently combined with current LIMS systems [24,27,28]. The development of LIMS did not stop after a hibernation period in the late 1990s. Current studies show that the critical constraints, such as low accuracy and precision in the elemental and isotope quantification, low mass resolution and considerable interference by abundant polyatomic species have been improved [1,25,26,29–31]. Furthermore, reduced mass spectrometric signal fluctuations, combined with high stability femtosecond laser ablation, pave the way to high-resolution chemical imaging and depth profiling analysis. The mass spectrometric analysis of these data allows for complex correlation analyses between isotopes and elements leading to improved quantification and identification of molecular parent compounds, and their location within the sample volume. These interesting new features can be of importance in analysing highly heterogeneous materials, including micrometre-sized mineralogical grains or identifying extant or fossilised microorganisms within rocks and soils [28,32–35].

The benefits of femtosecond LIMS are well-demonstrated in recent publications following numerous performance studies. Although fs-LIMS studies are still limited compared to extensive fs-LIBS [36–38] and fs-LA-ICP-MS [39] studies published in the last decade, this technique is rapidly developing and has the potential to compete with the latter techniques. Numerous advantages of femtosecond-pulse duration are known for both fs-LIBS and fs-LA-ICP-MS, and also for fs-LIMS which benefits from the resulting low ablation thresholds, absence of fractionation by vaporisation, improved spatial resolution for 3D

mapping applications, small ablated mass, and reduced sample damage. All these benefits originate from the physics of femtosecond ablation and subsequent plasma formation, which are described in Section 2. Section 3 introduces highly sensitive fs-LIMS systems used for a variety of studies on chemical composition in the last two decades. In Section 4, we summarise the analytical performance of fs-LIMS. Several applications of fs-LIMS are discussed in Section 5. Special attention is given to the comparison of analytical figures of merit (sensitivity, accuracy, and precision), as well as unique applications to the analysis of heterogeneous samples and micrometre-sized objects.

## 2. Laser-Matter Interaction in the Femtosecond Regime

### 2.1. Laser Ablation and Plasma Formation

This section summarises the major processes of laser ablation and plasma plume formation. Atomic ions, as analysed by mass spectrometry, form typically only a relatively small fraction of all other ablation products. Thus, to obtain the correct chemical composition of the material, one has to understand the relevant processes that support stoichiometric production of these ions, as well as any interfering processes contributing to fractionation effects. When these processes are understood and properly corrected, femtosecond laser ablation allows high-quality depth profiling and chemical imaging of the sample surface. The following section helps to understand the underlying processes that support these capabilities. We limit our discussion here mainly to processes involved during femtosecond laser ablation of solids.

Combined experimental and modelling studies have so far delivered a relatively detailed understanding of mechanisms of femtosecond laser ablation on various solid-state materials including metals, semiconductors, and insulators [37,39–43]. Contrary to ns-laser ablation, the fs-laser ablation mechanism can be described based on distinct timescales with clear temporal separation of different processes [43,44]. The overall process begins with the absorption of the laser energy by electrons of the sample material. The time duration of this process is determined by the pulse length of the laser. Sufficiently high deposited energy prompts ionisation within the laser spot. Femtosecond radiation exhibits extremely high peak powers ($>10^{13}$ W/cm$^2$) for moderate pulse energy ($>100$ nJ/pulse) due to its short pulse duration of ~100 fs. Practically, all materials can be ablated by focused fs-laser radiation. Independent of the pulse duration, laser radiation is absorbed by roughly the first 10 nm thick layer of material. However, femtosecond lasers have significantly lower ablation threshold energies than lasers with ns-pulse duration, e.g., for fs lasers, ablation of the sample already occurs at laser pulse energies of about 100 nJ/pulse, whereas several µJ/pulse (or more) is required for nanosecond lasers. Multi-pulse irradiation can further weaken the surface due to incubation effects. This decreases the ablation threshold energy and increases ablation efficiency for subsequent pulses [45,46]. Due to the short interaction with the sample, heat diffusion to surrounding regions of the processed area is significantly reduced. fs-lasers can efficiently heat electrons and generate a hot electron gas that is far from equilibrium with the lattice. The electron system gets heated to extremely high temperatures (~$10^4$ K), but electrons and ions cannot thermalize within the ultrashort pulse duration. The heat capacity of electrons is smaller compared to that of the material, therefore the electron system temperature rises to temperatures in $10^4$–$10^5$ K range, while ions and lattice are not heated up much in this time scale. It takes only a few hundred femtoseconds to a few picoseconds for the electron distribution to reach thermal equilibrium after femtosecond laser irradiation [47]. In contrast, the time to transfer energy from the electron subsystem to the lattice, which in turn induces thermalisation, is of the order of 1–100 ps (depending on the electron-phonon coupling strength of the material) and, thus, much longer [48]. Independent of the deposited energy density necessary for ablation, the material in the skin layer remains intact during the interaction time with a laser pulse. Therefore, even for stronger laser pulses, the perturbed material remains at a density close to that of the initial solid state during the interaction with the laser pulse. Conversely, even at elevated radiation intensities, the optical properties of the material

change rapidly during the radiation interaction due to a change in the electronic structure of the material [43]. These features distinguish fs-laser-matter interactions from those with ns-laser pulses. The laser pulse of about 100 fs duration is shorter than major relaxation times for electron-to-lattice energy transfer but also shorter than heat conduction and hydrodynamic expansion times. Hence, only a small fraction of the laser pulse energy is converted into heat. This results in non-thermal surface processing in the absence of melt and extended heat-affected zones (HAZ), which is frequently observed for longer-pulse laser ablations [49–51]. Thermal melting can still be observed as polarisation dependent periodic structures, and porous surface structures in lateral directions [52,53].

Two major processes responsible for absorption in metals and dielectrics involve the intra-band and inter-band transitions [43,44,54,55]. The first include the electrons excitation and heating in metals whereas the second is important for absorption in dielectrics. And involves single/multi-photon absorption and avalanche acceleration of electrons in valence band to the energy exceeding band gap. Dielectric subsequently can considered to be in the metal-like state [43]. Usually a few seed electrons are present in the valence band. Although the direct photon absorption is typically small, they gain energy by accelerations and collisions in laser field. Once they gain energy in excess of the band gap, can further collide with the electrons in the valence gap to transfer them for excitation into conduction band. Thus, the mechanism involves an avalanche of ionisation events. The other important ionization process that contributes into the total ionization rate is multi-photon ionization. This process is readily enhanced while fs-laser radiation is applied. Probability of simultaneous absorption of several photons (multi-photon absorption) is increasingly higher in case of femtosecond radiation than in case of ns-laser radiation [42,43,55,56]. Through multiphoton absorption induced by fs-lasers, a sufficiently large amount of energy can be deposited to cause ablation, even in materials that are transparent to the applied laser wavelength [56]. Typically, the light used needs to be of some sufficiently high enough energy (depending on the optical properties of the material) to be absorbed by the ablated material. Then the power density needs to be high enough to ionise all ablated material. For most materials ablated with ~100 fs laser pulses, laser intensity in the range from $10^{12}$ to $10^{13}$ W/cm$^2$ is applied [43]. At these conditions multiphoton absorption becomes important. The ionisation time duration (separation of electrons from the atoms) is becoming also shorter than the laser pulse duration. Ionization of the sample material already occurs early in the time-span of the radiation interaction (tens offs) [57] and can create a high-density plasma within the nm-thin surface. The primary process of electron excitation by the ultra-short laser pulse occurs in complete non-equilibrium conditions.

Since the laser energy is absorbed within the absorption length of material ($\approx$10 nm) and the thermal diffusion length is smaller than the absorption length, the electron temperature, plasma density, and pressure can become considerable high. For insulators, thermal energy electrons along with electrons produced by the multiphoton photoelectric mechanism are ejected out of the irradiated surface overcoming the attraction of the lattice. They induce an intense electric field ($\approx$10$^{11}$ V/m) between the irradiated surface and electron cloud, they form, existing up to 1 ps [58–60]. In this electrostatic field a number of ions can be pulled out of the surface via Coulomb explosion which has been found to contribute to the fast removal of several monolayers in dielectric materials [60]. In metals and semiconductors this process is less important. Free electrons in these materials are redistributed faster through diffusion and drift processes within material [61]. The heating rate of electrons by the laser radiation defines the excitation and ionisation rates. The typical time it takes for the electrons to heat the material to high temperature is several tens of femtoseconds, and this causes the conversion of any solid into plasma providing fs-laser irradiance reaches power densities $10^{12}$ W/cm$^2$. The hot electrons in the irradiated spot thermalise with the ions and lattice by electron–phonon coupling within 10–100 ps which leads to the temperature increase in a small volume. The rate of energy deposition is so fast that the material cannot evaporate continuously, and extremely high temperature and pressure is built up beneath the evaporating surface. What follows is the breakdown

of the irradiated volume [62], ejection of atomic clusters by Coulomb explosion [63], and the onset of plasma from the ablation plume [64]. Due to fast and isochoric heating, a rapid temperature increase to up to 10,000 K occurs, followed by an adiabatic expansion of the plasma volume [65]. While expanding, the plasma plume cools down through adiabatic expansion [64]. The laser-induced plasma experiences different stages during the expansion that provide ways for particle generation from the vapour phase [66]. The whole ablation process may last up to several microseconds. This second ablation phase is due to a phase explosion after heterogeneous or homogeneous nucleation. As such, it takes a significant amount of time for the nucleation bubbles to develop and grow, explaining the temporal delay. Several intermediate thermodynamic pathways are observed with a decrease in pulse energy and the energy gradient within the irradiated volume, including vaporisation, fragmentation, phase explosion and spallation [40,41,67–70]. The absorption of nanosecond laser radiation ($\sim 10^8$–$10^9$ W/cm$^2$) is weak and dependent on the chemical and physical properties of the material. The radiation is absorbed typically via the inter-band transitions, defects, and excitations. The electron-to-lattice energy exchange time in a long-pulse ablation mode is several orders of magnitude shorter than the pulse duration. As a result, the pulse energy is partially buffered by plasma–light interactions and the major part of the remaining energy is consumed by heat diffusion [38,49]. For this reason, the electrons and ions are in equilibrium and the ablation is dominated by thermal evaporation. In ns-laser ablation, melting becomes the dominant process and ablation occurs via hydrodynamic sputtering from Rayleigh–Taylor and Kelvin–Helmholtz shear instabilities at the melt pool surface [71,72]. Figure 1, panel (a) displays schematically phase transitions occurring during sample irradiation by ns-laser with final vaporisation of the sample and formation of the plasma. Figure 1, panel (b) displays schematically two-step ablation mechanism involving an initial formation of hot electrons and cold sample array followed by electron-heating and plasma explosion. Figure 1, panel (c) and (d) illustrates structure of the plasma plume produced by irradiation of the surface by ns- and fs-lasers.

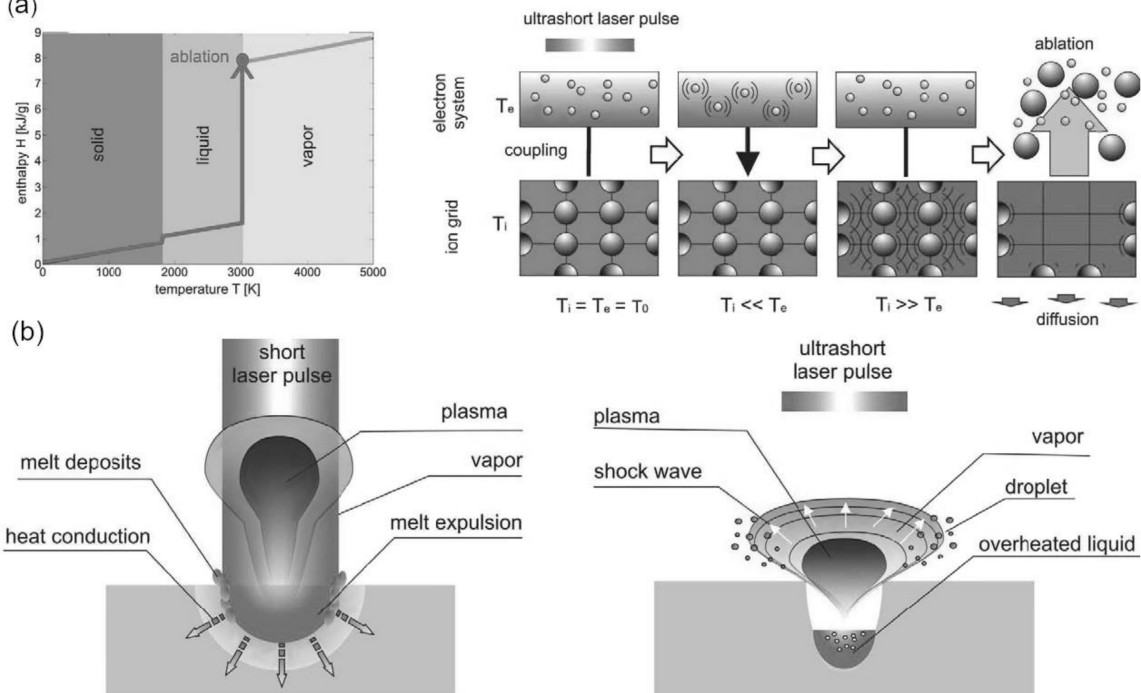

**Figure 1.** Ablation models: (**a**) classical ablation model; (**b**) two-temperature model as basis of ultrafast ablation model. Beam-Matter interaction: (**c**) classical beam-matter interaction; (**d**) ultrafast beam-matter interaction (Reprinted with permission from [73]).

Multiphoton absorption processes of fs-laser photons affect the ablated spot size. Typically, the size of the focused laser spot is defined by the diffraction limits, which also limits the lateral resolution in laser ablation. However, the ablation spot is typically observed to be smaller than the expected laser focal spot size. Due to non-linear absorption, the absorbed energy distribution becomes narrower with increasing order (n) of multiphoton absorption. The effective absorption coefficient for n-photon absorption is proportional to the n-th power of the laser intensity, $I$ ($I^n$) and the effective laser spot size for n-photon absorption becomes respectively smaller. The effective beam size d for n-photon absorption is expressed as $d = d_0/n^{1/2}$ where $d_0$ is the actual spot size of the focused laser beam [74,75].

Ablation at the surface, in addition to atoms and atomic ions, also produces larger particles, which can further catalyse the particle formation rate in the expanding plasma plume. Non-uniform power densities are at the onset of various processes dependent on the incident laser intensity. Thus, by applying laser radiation with a Gaussian intensity-profile, these processes get mixed. Destruction of the sample integrity (appearance of cracks, flaking of the surface), melting, ablation, and ionisation due to the spatial intensity distribution across the focal spot take place across the laser-affected area even if the average spatial intensity is higher than the ablation threshold. Some improvements in the reduction of particles and better control of the ablation products can be made when local fluences are above the ablation threshold and equal at every point across the focal area. Application of laser radiation with a flat-top intensity-profile offers more optimal conditions for the ablation than typically applied Gaussian-shaped beam, which exhibits a changing intensity across the ablation area [43]. So far, the flat-top fs-laser beam is not used, and mostly Gaussian laser beams were applied in LIMS experiments [76].

### 2.2. Diagnostics of the Ion Formation in fs-Laser Ablation

Femtosecond LIMS studies on the dependence of the ion formation efficiency (ion yield) and the kinetic energy distributions are limited in number but more are expected in the near future. The ion yields produced in laser ablation are a complex function of laser ablation ion source parameters and are typically referred to the ionization efficiency, the fraction of ionized atoms in the ablated plume. Early studies by Hergenröder et al. [18] compared ns-and fs- induced ablation of brass samples indicating the presence of several ion speed components upon both types of ablations. Similar observations were made in fs-laser ablation of gold and copper samples by Amoruso et al. [77], although the ratio of the high-/low- speed components differed from the ratios obtained by Hergenröder et al. [18]. In later diagnostic studies by Amoruso et al., fs-ablation of Cu, Al, Mg, Cr, Fe, and W samples applying laser fluence in the range from about 0.5 J/cm² up to 75 J/cm² was conducted with three diagnostic techniques including a Faraday Cup (FC), a Langmuir Probe (LP) and imaging by ICCD camera. Up to a laser fluence of 10 J/cm², the ion angular distributions were found to be well described by the Anisimov expansion model [78]. At larger fluences, a broadening of the angular distribution of ions was observed. The light metal ions (Al, Mg) generally showed a narrow, forward-peaked velocity distribution, and high peak ion yield compared to heavier metals (Fe, W, Cu). In this study, ion yields were also investigated as a function of laser fluence (J/cm²) [79]. First, a clear spatial splitting of neutrals and ions, with the charged plume component expanding ahead of the neutral population, was observed with an increased relative abundance of atomic ions at higher fluences. Three different ion emission regimes were identified as a function of the laser fluence. Past the ablation threshold (~0.5 J/cm²), the first transition was observed at about 5–6 J/cm², with a maximum in the specific ion yield, i.e., the yield per unit fluence, and a second transition was observed at a high fluence of 50 J/cm² (Figure 2). For laser fluences <50 mJ/cm², the experimental results were in the agreement with the two-temperature model description of the fs laser-sample interaction and subsequent material decomposition mechanisms. However, for larger laser fluences, another complex mechanism was inferred [80]. These quantitative studies show that in typical LIMS experiments at a laser fluence of ~1 J/cm², one would expect ion yields of about ~$10^{10}$ ions/cm² for metallic

and for high laser fluence instruments approximately 100 larger. Furthermore, the studies showed that the ion yield depends on the volatility of the elemental metallic samples and lesser dependence on IP (Table 1).

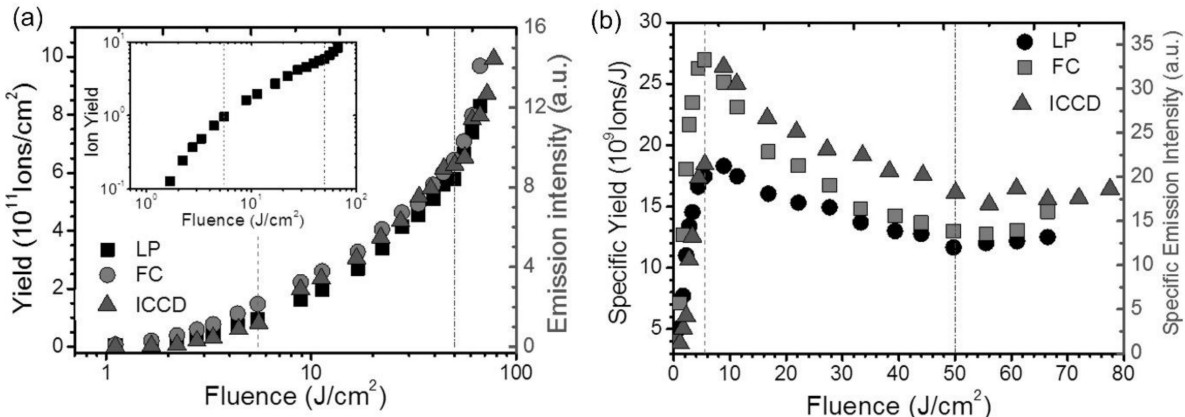

**Figure 2.** Copper ion emission yield (**a**) and specific ion yield (**b**) as a function of the fs-laser fluence. The two vertical lines mark the transition between different regimes. The inset in panel (a) shows the Langmuir probe (LP) data on a log-log plot. The left axis represents the LP and Faraday Cup (FC) measurements and the right axis represents the spectrally resolved intensified charged coupled device (ICCD) imaging. Reprinted with permission from [79].

**Table 1.** Physical properties of the selected metals: A, atomic weight; $T_m$, melting point; R, reflectivity at 800 nm, normal incidence; IP, ionisation potential. $Y_0$ is the peak value of the ion yield registered along the direction normal to the target surface [79].

| Element | A (u) | $T_m$ (K) | R (%) | IP (eV) | $Y_0$ ($10^{11}$ions/cm$^2$) |
|---------|-------|-----------|-------|---------|------------------------------|
| Mg | 24.30 | 923 | . . . | 7.65 | 5.7 |
| Al | 26.98 | 933 | 86.8 | 5.99 | 4.9 |
| Cr | 51.99 | 2180 | 56.8 | 6.77 | 3.8 |
| Fe | 55.84 | 1811 | 56.12 | 7.90 | 4.5 |
| Cu | 63.54 | 1357 | 96.3 | 7.73 | 1.9 |
| W | 183.84 | 3695 | 49.6 | 7.86 | 2.4 |

Overall, the values of the ionisation efficiency (fraction of atoms ionised in the ablation plume) in nanosecond LIMS was summarised in the review with the ion yields in high laser fluence to lie in the interval from 10% to 80% (including multiple ions) [3]. For lower laser fluences above the laser ablation threshold, the ionisation efficiency is of the order of $10^{-3}$ (Figure 2). So far, there are no systematic fs-ablation LIMS studies of ion yield as a function of wavelength. For ns-ablation, large differences between the ionisation yields dependent on wavelength and material are observed (large matrix effects). Due to the nonlinear absorption of fs-radiation, the dependence of ablation efficiency and ion generation efficiency on material and wavelength is observed to be smaller than that of ns-lasers but still exists. While conducting the UV fs-laser ablation again less dependence on material properties and smaller fractionation effects compared to the IR fs-laser ablation can be observed [81,82].

The comparison of ion yields obtained in high irradiance ns- and fs-laser ablation (laser wavelength 1064 and 1023 nm, respectively) were reported by the Huang group [20]. These extensive studies were conducted on 29 solid samples with 10 different matrices, including six metals and four insulators using buffer-gas-assisted LI-TOFMS. The mass spectrometric analysis led to conclusions that for most of the elements, except for the nonmetallic elements, the ionisation yields are proportional to atomic abundance in the bulk material for both ns- and fs- ablation. Nevertheless, the ion yields produced by ns-ablation undergo larger fluctuations compared to fs-ablation ion yields. The ion yield dependence

on the material was observed to be less for fs-ablation due to lack of thermal diffusion, production of a dense plasma and more stable ionisation. The remaining differences in the stoichiometric ionisation were accounted for in the different ionisation potentials and ionisation cross-sections for different elements.

### 2.3. Double Pulse fs-Laser Ablation and Ion Yields

The characteristics of fs laser pulses permit applications of more complex irradiation schemes such as short delay pulse sequences or tailored pulse shapes. One of the arrangements, known as double pulse fs-ablation, was particularly theoretically and experimentally particularly well-explored due to its application in material processing and synthesis [83–85], chemical analysis by LIBS [86], and was recently applied in fs-LIMS [82,87,88]. Femtosecond double pulse (DP) irradiation offers better control over ionisation yields and polyatomic species produced in laser ablation [87,88]. This control is achieved by tuning the laser pulse energy of each laser pulse and inter-pulse delays (ps range) [85].

The mechanisms of femtosecond double-pulse laser ablation were investigated by modelling (e.g., hydrodynamic simulation, molecular dynamics) [89,90] and by plasma diagnostics [91–93]. In hydrodynamic simulation methods, one can identify a pressure/density wave produced by the first laser pulse that propagates into the sample. Behind this wave, a tensile stress wave is formed and propagated through the liquid layer which results in the mechanical fragmentation and ablation of the layer. When the delay is much shorter than the electron-ion relaxation time, only these two waves appear. When the delay is on the order of the relaxation time, the second pulse creates the second wave, thus reducing the intensity of the first tensile wave which also causes the depth of the ablation crater to decrease. For delays longer than the electron-ion relaxation time, the second pulse reheats and decelerates the ablated material, and the ablation crater is then formed by the first pulse only with the crater depth smaller than that of the single pulse ablation alone [90]. The results of this study show that ablation can be suppressed due to the formation of the second shockwave.

In a LIMS application, femtosecond double-pulse laser ablation can improve the quantitative chemical analysis and further improve the resolution of chemical depth profiling [82,87,88]. In addition, at delays in a range of tens of ps, DP leads to a significant reduction of polyatomic ions, by reheating produced plasma and ionising neutral gas, which reduces isobaric interferences and an increase of ion yields up to a factor of 10–50 compared to single pulse femtosecond ablation [82,87,88].

### 3. LIMS System: fs-Laser Ablation Ion Source and Time-of-Flight Mass Analyser

fs-LIMS is built with two components, namely an fs-laser ablation ion source and a mass analyser. In the present discussion, we take only TOFMS into account since it is an obvious choice for pulsed lasers. The fs-laser radiation source consists of an fs-laser system and an optical system shaping and focusing laser pulses on a sample. The ions produced in the ablation process are analysed either instantaneously by TOFMS or after additional pretreatment [21,24]. In one of the high fluence LIMS systems, the ions are first thermalised by passing them through a gas collision cell. After that cell, an additional ion optical lens system is used to transport the ions to the ion extraction region where they are introduced into the mass analyser via the orthogonal extraction method [22,23]. Frequently, ion-optical systems of current TOFMS used for fs-LIMS undergo careful ion-optical modelling to optimise their ion transmission and mass resolution. This is an important procedure in obtaining high-performance miniature TOMS [21] and advanced TOFMS [24] capable of supporting the high mass resolution.

### 3.1. Femtosecond Laser Radiation

Developments of laser ablation ion source and application of femtosecond-laser radiation for ablation considerably improved LIMS performance. Only in the last two decades with the appearance of commercially available fs-lasers, and the improvements of their

long-term stability were fs-lasers made an applicable technology to a wider audience than just laser physicists. Currently, these systems can be operated without frequent optical alignment producing temporally stable fs-radiation. Nevertheless, for faster progress in some applications based on fs-lasers, the price of these lasers and accompanying fs-optics are still limiting factors.

The first generation of femtosecond laser pulses was demonstrated in the 1970s, and the invention of the Ti:Sapphire laser in the 1980s paved the way for broader interest in this technology, followed by the invention of the chirped-pulse amplification technique (CPA) in 1985 [94]. This led to the construction of high-power femtosecond lasers and their wide applications in various fields due to the commercialisation of these instruments [95–97]. From the large variety of fs-lasers developed so far, only a few of them have been made commercially available [98]. Despite the relatively simple operation of the commercial fs-laser system, their maintenance can be still extensive and expensive. It also requires qualified personnel to support the continuous and stable operation of this system. This is likely one of the current constraints limiting the broader application of fs-LIMS, which however is expected to diminish with further improvement of fs-lasers and beam delivery means (e.g., with wider usage of fibre lasers) [99]. Although the fs-laser system is relatively complex, it is not complicated. Here, we introduce the basics of fs-laser technology, which is applied in the current fs-LIMS. The controllable operation of the fs-laser is an important factor and contributes to the success of the fs-laser ablation ion source. The current commercial fs-laser systems use a gain medium self-mode-locking (or Kerr lens mode-locking) based on the spatial phase-modulation effect, which favours coherent short-pulse emission [100] and the chirped-pulse-amplification (CPA) method (Figure 3) [94].

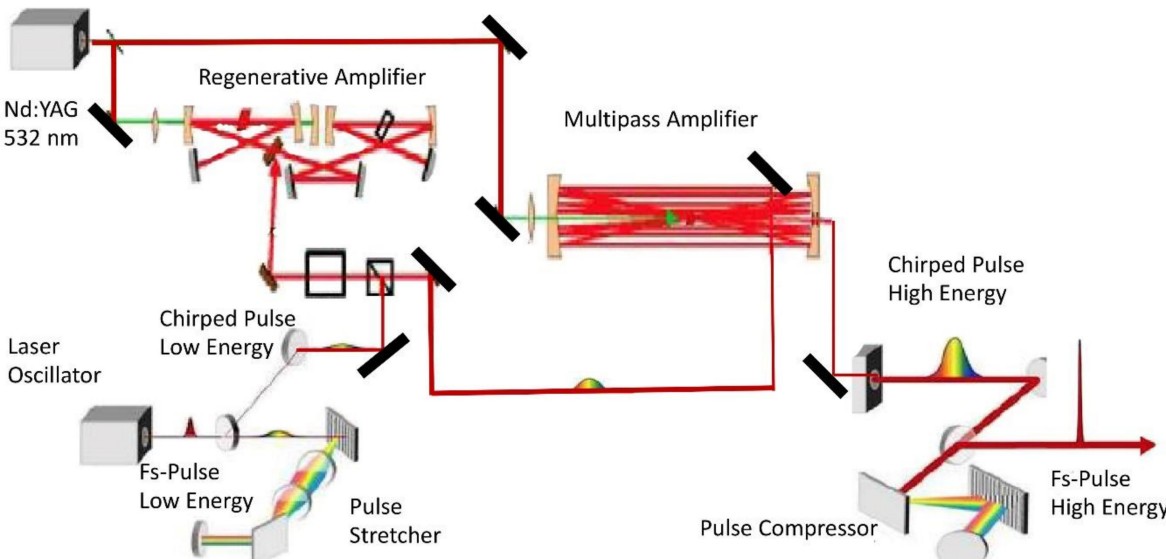

**Figure 3.** Functional schematic diagram of an fs-laser system used typically in fs-LIMS (see text for more details).

Here we outline only shortly the underlying principles of fs-laser operation and describe the functionality of fs-laser components based on the dedicated monographs e.g., [98,101,102]. The fs-laser consists of a seed laser and amplification stages. The seed laser produces a low-energy (1 nJ) fs-pulse train of up to 80 MHz repetition rate using the mode-locking technique. In commercial applications, Ti:Sapphire crystals are commonly used as a lasing medium [101]. The TEM00 seed beam is coupled into a single-mode fibre where the pulses undergo self-phase modulation and group velocity dispersion and become linearly chirped. After transformations in the fibre, the beam is sent through a dispersive delay line formed by two diffraction gratings, causing spatial dispersion of different wavelengths and their re-collimation. The optical path through the grating pair is longer for the long-wavelength than for the short ones. This system introduces a negative

group velocity dispersion, which can be used to compensate for the positive chirp of the self-phase-modulated pulse. With a pair of retroreflector mirrors, the beam is reflected into the grating pair at a different height. The spatial chirp on the beam is cancelled while doubling the dispersion of the system. When a transform-limited pulse is sent through the compressor, the output pulse is negatively chirped with the front of the pulse bluer than the tail and its duration increased. Suitable arrangements can introduce positive as well as negative group delay dispersion. When introducing negative group delay dispersion, the corresponding device is termed a compressor, while a device introducing positive group delay dispersion is termed a stretcher.

The output pulse energy of the femtosecond oscillator (~1 nJ per pulse) is low. Additional amplification of these pulses is provided by amplification stages. Two techniques, multipass and regenerative amplification, are typically employed. A second stage, typically an Nd:YAG pumped Ti:Sapphire crystal delivers the necessary energy via population inversion, which is used to amplify the low-energy pulses. The ns-pulse generated in the amplifier is energy efficiently transferred to the fs-pulse. The fs pulse must be chirped to several picoseconds. This pulse prolongation is achieved in the stretcher containing a grating that spatially disperses the fs-pulse $10^3$–$10^4$ times before injection into the amplifier. After the amplification process, the pulses have to be recompressed, compensating for additional phases accumulated during the amplification process [97,98]. Due to a high saturation fluence that typically occurs in solid-state materials, only a small optical gain per pass can be achieved in the amplifier. To extract the energy stored in the medium, one must use multiple passes in the amplifier. Multi-pass amplification is based on a bow-tie type of amplifier, in which the different passes in the amplifier are separated geometrically. In regenerative amplification, a pulse is trapped in a laser resonator until it has extracted all the energy stored in the amplification medium, regardless of the gain per pass. Trapping and dumping the pulse is done using a Pockels cell and a broadband polariser. The Pockels cell is initially set to be equivalent to a quarter-wave plate. When a pulse is in the resonator, the voltage on the Pockels-cell crystal is switched so that it becomes equivalent to a half-wave plate. The pulse stays in the cavity until it reaches saturation, and a second voltage step is applied to the Pockels cell to extract the pulse. Because the regenerative amplifier operates at 1 kHz, and the seed laser at MHz repetition rates, the pulses have to be synchronised. This is accomplished by a radiofrequency (RF) divider. The MHz signal from the seed laser is divided down to a rate that can be sustained by the Nd:YAG laser. After amplification, the pulse must be compressed again in the temporal compressor functioning as a reversed stretcher. Most of the current commercial fs-laser systems are Ti:Sapphire systems with fundamental wavelength output at 775 or 800 nm. Table 2 summarises the characteristics of few femtosecond lasers used so far in LIMS instruments.

**Table 2.** Femtosecond laser systems applied in LIMS.

|  | Ti:Sapphire [19] (Spectra-Physics) | Ti:Sapphire [21] (Clark-MXR, Inc.) | Ytterbium-Doped Semiconductor Laser S-Pulse, Amplitude Systems [20] |
|---|---|---|---|
| wavelength | 800 nm | 775 nm SHG, THG | 1030 nm |
| duration | 45 fs (effect. 75 fs) | 180 fs | 500 fs |
| repetition | 1 kHz | 1 kHz | 10 Hz |
| output, max. | 2 mJ | 1.2 mJ | 1 mJ |
| Spot dia. | 8 μm | 5–40 μm | 40 μm |

More advanced temporally and intensity modified laser pulses (pulse-shaping method) can help to control ablation products and ionisation efficiency [90]. Using arbitrary pulse sequences (typically at the expense of temporal spread), one can achieve the required optimisation and control over the plasma parameters [103,104]. Temporal tailoring of femtosecond laser pulses by adaptive feedback loops was demonstrated in material sci-

ence [103]. This method can enable better optimisation of laser interaction with the sample by adjusting the delivered energy.

In the application of fs-radiation for fs-LIMS, one has to consider its several effects on the ablation efficiency. The chirped pulse amplification (CPA) technique often produces pre-pulses containing a significant amount of energy [94]. A high contrast ratio of 100 or more is required between the energy of the main pulse and any pre-pulses to make the effects of pre-pulses on ablation negligible [105]. An additional source of a pre-pulse is amplified spontaneous emission (ASE) from the laser amplifiers. To eliminate ASE, the laser design typically includes successive stages of amplification interspersed with spatial filters.

Another important effect produced by fs-radiation is filamentation, which is the consequence of self-focusing of the laser beam while travelling from the laser output. The high-intensity central part of the wave front would see a higher index of refraction in the transfer medium than the low-intensity edge due to a nonlinear increase of the index of refraction. Thus, the light wave propagates slower through the central part of the lens than through its edges, resulting in the curvature of the wave front toward the axis of propagation. Sufficiently intense femtosecond laser pulses undergoing self-focusing in any transparent optical medium (including air) will collapse to the generated plasma. During filamentation of a femtosecond laser pulse, some fundamental nonlinear processes are involved in the propagation and interaction, including nonlinear pulse phase change [106]. A recent review on fs-LIBS discussed filamentation effects in the context of this spectroscopic method [38].

### 3.1.1. Laser Pulse Shaping

More advanced temporally and intensity modified laser pulses (pulse-shaping method) can help to control ablation products and ionisation efficiency [90]. Using arbitrary pulse sequences (typically at the expense of temporal spread), one can achieve the required optimisation and control over the plasma parameters [103,104]. Temporal tailoring of femtosecond laser pulses by adaptive feedback loops was demonstrated in material science to enable better optimisation of laser interaction with the sample by adjusting energy delivery [103]. Temporally or spatially shaping femtosecond pulses allows the control of localized transient electron dynamics and material properties to adjust material phase change [107].

Typically, it is difficult to determine the thickness of an ablated volume achieved by a Gaussian beam because they do not produce flat bottoms and reveal conical craters with temporally nonlinear ablation behaviour. A flat-top laser beam profile proves to be necessary for such an application. Such laser pulse yields a homogenised ablation rate over the laser spot, so that flat crater bottoms can be achieved, forming cylindrical craters. Ablation rates approaching <1 nm per shot, significantly steeper crater walls, and minimal surface damage in comparison to the Gaussian craters were observed in recently reported studies [108]. Flat bottom near-cylindrical and "splash-like" conical crater geometries observed in these experiments also indicated different ablation regimes for both cases despite using similar laser energies [108].

In recent LIMS experiments, the laser ablation optical system is further modified to accommodate a second laser pulse. To support their flexible delay, interferometric setups (e.g., Mach-Zehnder) and optical elements allowing laser pulse intensity tuning) are included [88]. Such systems, called double pulse (DP) systems, are found to be very useful for increasing ionisation efficiency and control over the abundance of polyatomic/molecular species produced during the ablation [88]. In a very recent investigation, this system was extended to UV femtosecond radiation range [82]. Both pulses were then introduced into an SHG/THG [82]. UV femtosecond radiation with up to 15–20 μJ per pulse can be produced from an initial fundamental pulse energy of 1mJ and tuned with specially designed polarisation optics.



To achieve a high-energy density at the sample surface, one should minimise both the energy losses along the transport path and in the focal volume at the sampling point. The major obstacle for a laser beam to propagate a long distance is self-focusing caused by the intrinsic non-linearity in a medium. Self-focusing before the focus may result in the optical breakdown at certain power of a beam and therefore prevents the laser beam from delivering the energy into the desired focal area. The power of a laser beam, wavelength $\lambda$, and intensity, I, aimed to deliver the energy to a desirable spot inside the bulk of a transparent solid, should be kept lower than the self-focusing threshold for the medium. By using high-aperture optics, one can obtain the intensities above $10^{13}$ W/cm$^2$ in the focal plane and stay below the self-focusing threshold [43].

### 3.1.2. Generation of fs-Laser Radiation with Shorter Wavelength

Higher harmonics of the output frequency can be obtained by using nonlinear quadratic crystals such as BBO ($\beta$-barium borate), LBO (lithium triborate), or KTP (KTiOPO$_4$) [109,110]. In a crystal medium, two incident fields at frequencies $\omega_1$ and $\omega_2$ can produce new radiation at $\omega_3 = \omega_1 + \omega_2$ or $\omega_3 = \omega_1 - \omega_2$. The output of the fs-laser can be used as a fundamental wave for second-harmonic generation, frequency mixing, and parametric amplification. The latter two methods are used to produce tunable infrared or visible fs-radiation (optical parametric generator, OPG, non-collinear optical parametric amplifier, NOPA) [111]. A simpler way to produce UV fs-radiation is by combining the output of the second-harmonic with the fundamental wave on a third-harmonic crystal (THG). The ideal crystal for this application must have a small group velocity difference between the fundamental and harmonic waves and a large nonlinearity and must exhibit no absorption at both the fundamental and the harmonic wavelengths. The crystal should also be transparent at twice the harmonic frequency to avoid two-photon absorption. A thin BBO crystal is typically used for the second-and third-harmonic generation and has 40–50% and up to 10% conversion efficiency, respectively [112,113].

### 3.1.3. Femtosecond Laser Optics

Refractive optics (i.e., a lens) are a common way to focus light on a spot and the lenses are widely used in the setup for laser ablation source. To achieve a focal spot size of about 10 μm or less, a telescope system is often used to expand the beam to the required diameter before focusing [21,114]. Nevertheless, refractive optics are subjected to chromatic and spatial aberrations, which typically increase the complexity of the optical design. Femtosecond-laser radiation is more sensitive to employed optical elements compared to other lasers with longer pulse widths. The spectral width of a 100 fs pulse ($10^{-13}$ s) corresponds to about 10 nm in wavelength ($\Delta\nu = 4.41 * 10^{12}$ Hz) and can suffer from group velocity dispersion [101]. IR femtosecond pulses suffer less from group velocity dispersion compared to the UV pulses while propagating through thick transparent materials. Dispersion effect also takes place when light is reflected from a multi-layered dielectric mirror, such as those used in laser cavities. Reflection of light from such mirrors is the result of constructive interference between rays that have been partially reflected at the interfaces of $\lambda/4$ dielectric layers. Inherently, group velocity dispersion occurs in these mirrors and can affect the overall beam profile. The wide spectral width of femtosecond optical pulses requires a large enough bandwidth for every optical element used in an experimental setup to accommodate all the frequency components of the applied pulses. Femtosecond radiation is a delicate entity, such that any spectral narrowing induces a broadening in the time domain that can be critical to the ablation process. To avoid spurious phase distortions and dispersion while transferring femtosecond radiation from one point to another, aluminium- or silver-protected broad band mirrors are recommended. Moreover, reflective optics with aluminium- or silver-protected spherical or parabolic mirrors are often more advantageous for femtosecond radiation focusing over refractive optical elements. By applying such elements, high-quality focusing spots without aberration or astigmatism can be obtained [24].

### 3.1.4. Optical Arrangement of Laser Ablation Ion Source

Optical focusing of laser radiation in LIMS was realised so far by a single lens, a doublet lens system, both combined with a telescope, and by Schwarzschild optics also combined with a telescope [15,16,21,24,115]. The ablation spots sizes produced by a single lens-based focusing system are several tens of μm in diameter. With a telescope expanding the laser beam to 20–40 mm and focusing it by a doublet lens, one can focus a laser beam to a spot of 10 μm or less. Particularly convenient in femtosecond, LIMS application are Schwarzshild optics [24]. Two spherical mirrors are placed concentrically with each other and separated by twice their focal length. Laser radiation is reflected from the outer sphere of the first mirror onto the inner sphere of the second mirror which subsequently focuses the radiation on the sample surface In the Schwarzschild configuration, third-order spherical aberration such as coma and astigmatism can be eliminated, resulting in optimal focusing conditions [116]. By focusing, IR femtosecond radiation laser can be focused to the spot size of 1–2 μm [15,24,117]. The design of the optical systems for required laser focusing conditions can be optimised using e.g., the ZEMAX software [118]. The Schwarzschild laser focusing system allows for tight focusing conditions with minimal aberration at the focal point. Schwarzschild optics were implemented in early nanosecond-LIMS optical setups [15] and recently in LMS-GT [15,24].

In recent LIMS experiments, the laser ablation optical system is further modified to accommodate the second laser pulse. This system is called a double pulse (DP) and is found to be very useful for increasing ionisation efficiency and reduction of the abundances of polyatomic species produced during the ablation [88]. The DP system splits the femtosecond laser beam into two beams, using a 50% beam splitter. One of these laser beams is delayed against the other, by passing it through a movable retro-reflector positioned on a 300 mm long remote-controlled linear delay stage. The two beams are combined finally again using another beam splitter, resulting in a train of two laser pulses with adjustable delay [88]. In a very recent investigation, this system was extended to UV femtosecond radiation range [82]. In the optical setup, first, the fundamental output was split on two pulses of equal energy, followed by an arrangement of their delay using an interferometer. These pulses were introduced into SHG and THG units [82]. UV femtosecond radiation with up to 15–20 μJ per pulse can be produced from the initial fundamental pulse energy of 1 mJ and tuned with specially designed polarisation optics.

To create the high-energy density in the bulk of the material, one has to minimise both the energy losses along the transport path and in the focal volume at the sampling point. The major obstacle for a laser beam to propagate a long distance is self-focusing caused by the intrinsic non-linearity in a medium. The intensity *I* of a laser beam, at a given wavelength λ, should be kept lower than the self-focusing threshold for the medium, to deliver the desired energy density to the spot of analysis. Using wide beams in high-aperture optics, one can obtain the intensities above $10^{13}$ W/cm$^2$ in the focal plane and stay below the self-focusing threshold during beam transport [43].

Recent studies show that fs-laser can be readily focused on the spot size beyond the diffraction limit and make fs-LIMS truly nano-probe [119]. The method is based on near-field effect. By placing an aperture or a tip in the vicinity of the sample surface, near-field localised enhancement of laser energy can be achieved. The investigation conducted by Liang et al. used LIMS with a modified STM, where a silver tip-enhanced ablation of 10 nm thick Ti layer coated on the Au substrate was achieved. Craters of 50 to 80 nm in diameter were formed by applying fs-laser (515 nm, 500 fs). The laser fluence was sufficient to ablate, atomise and ionise the sample.

### 3.2. TOF Mass Analyser

The principles underlying TOFMS instruments can be found in recent reviews [120,121]. Several other reviews cover perspectives and historical developments of TOFMS [122–124]. TOFMS, contrary to earlier predictions, has become an important mass spectrometric technique [121]. The mass separation of TOFMS occurs because the ions of different mass

and same kinetic energy have different velocities and, after sufficiently long flight time, arrive at the ion detector in well-separated ion packets according to their mass. Using relatively modest accelerating potentials, ion flight times are on the order of 10s to 100s of microseconds allowing fast measurements. Because the mass range of TOFMS is defined by the duration of the experiment, theoretically, TOFMS can have an unlimited mass range. A complete mass spectrum is generated with each laser shot, which is a unique property of TOF instruments, compared to scanning mass spectrometers. Therefore, rapid chemical analysis with excellent statistics can be conducted with the current ion source repetition rate of kHz. TOFMS couples very well with pulsed ion sources such as femtosecond laser ablation ion source [125]. The signal-to-noise (S/N) of a TOFMS increases with mass resolution, because of better time focusing, where it decreases with increasing mass resolution for quadrupole and sector instruments.

The mass resolution and ion transmission in TOFMS instruments have improved considerably since the first demonstration of the TOF concept. Almost all current TOFMS uses an ion mirror (reflectron) significantly improving mass resolution [126]. High ion transmittance is achieved using grid-less ion optical elements instead of, e.g., ion mirrors with grids [127]. The ion optic voltage settings are derived from the ion trajectory modelling. Thus, the ion optics are optimised to confine, accelerate and focus the ions, ideally without transmission losses, on their trajectory to the ion detector. In the current femtosecond-LIMS instruments, the TOFMS flight tube axis is positioned either orthogonal or collinear with the direction of ion velocities. In an optimised collinear configuration (see also Section 3.3.2), all ions produced in ablation can be introduced into the mass analyser for analysis [24].

The main shortcoming of all linear TOFs, having a straight path from the ion source to the detector, is their inability to compensate for the inhomogeneities in the ion kinetic energy (velocity) ranging from 1 eV up to 1 keV, depending on laser fluence [128]. Under conditions of acceleration through static electric fields, compensation for initial velocity distribution is unavoidable and further dependent on $m/z$. Nevertheless, by arranging the region of ion production just at the entrance into the mass analyser and applying sufficiently large extraction fields, a high degree of compensation to ion angular and kinetic energy spreads can be achieved. With a sudden acceleration, the extraction conditions similar to that of the pulsed acceleration field can be achieved. Detailed ion-optical settings can be further optimised to obtain desired mass resolution and ion transmission by applying ion trajectory modelling. Orthogonal extraction (OA), as applied in the Huang group, increases the mass resolution, dynamic range, and duty cycle of TOFMS coupled with temporally broad or continuous beams (see also Section 3.3.1) [22,24,25,129]. Concerning TOF flight tube, ions flying in an orthogonal direction can be extracted from a source continuously and periodically accelerated by a pulsed electric field into the interior of the mass analyser in discrete ion packets (see also Section 3.3.1). However, from a temporally broad ion beam arriving at the ion extraction region of the OA, only part of the ion packet is directed into the interior of the mass analyser by an ion repeller. This can cause ambiguities in the ion selection. Nevertheless, in orthogonal acceleration, the initial ion kinetic energy distribution of the ions is converted into narrow ion velocity distribution in the ion extraction direction. Ions in an orthogonal acceleration experiment acquire kinetic energy approximately a hundred times higher than their initial kinetic energies and, consequently, time dispersion of the ion bunches can be significantly reduced. Orthogonal acceleration increases the mass resolution of heavy ions; the resolution of light ions decreases as the square root of the mass-to-charge value [130]. While the initial velocity spread in the orthogonal direction is minimised, there is a spatial distribution of ions across the detector surface. Typically, an orthogonal accelerator has a system of grids, which results in a decrease in ion transmission. The repetition time cannot be less than the finite flight-time of the heaviest ion; otherwise, light ions from one cycle may be registered before heavy ions from the previous cycle.

The mass resolution derived from the mass spectra is affected by the temporal width of the ion packets arriving at the detector, $\Delta t$, which has three major contributions: $\Delta t_{tof}$, a spread in ion source and flight through the mass analyser; the spread due to the ion

detection and coupling to the transmission line system $\Delta t_{det}$, and the finite time resolution of the acquisition system ($\Delta t_{acq}$). They affect mass resolution according to the equation:

$$\mathbf{R} = \frac{\mathbf{m}}{\mathbf{\Delta m}} = \frac{\mathbf{t}}{\mathbf{2\Delta t}} = \frac{1}{2}\frac{\mathbf{t}}{\mathbf{\Delta t_{tof} + \Delta t_{det} + \Delta t_{acq}}} \tag{1}$$

where t is the time of flight measured at the peak's maximum and $\Delta t$ is its full width at half maximum (FWHM). The pulse width ($\Delta t$) of a particular atomic (isotope) ion package is determined in the measurement. While the ion packet spread by the ion optical settings is proportional to the time of flight t, the other terms are constant. The contribution of $\Delta t_{acq}$ is typically determined from the data acquisition system specification, while $\Delta t_{det}$ can be obtained by measuring single ion events and $\Delta t_{tof}$ is dependent on the ion-optical design of the instrument. All three terms should be minimised during the design process of the instrument to maximise the mass resolution. For most LIMS systems, $\Delta t_{tof}$ is much larger than other contributions. Recently developed high-resolution LIMS $\Delta t_{tof}$ becomes comparable with other terms and improvements of the other spreads are required [24].

### 3.3. Selected Femtosecond-LIMS Systems

The historic development of LIMS instrumentation is well-documented in a recent review and includes a brief overview of the first-generation instruments and more recent designs developed between 1960 and 1990 [1]. The limited number of femtosecond-LIMS instruments so far is a result of the high price of the fs-laser system and perhaps of the prevailing experience from the nanosecond predecessor of this method about the qualitative rather than quantitative capabilities of LIMS. However, already early femtosecond LIMS indicated many improved capabilities of this technique. A TOFMS equipped with a commercial Ti:Sapphire femtosecond-laser (75 fs, 800 nm, 1 kHz) and a coaxial ion extraction system, a two-stage reflectron and microscope system, demonstrated high capabilities for depth profiling analysis [19]. In this instrument, focused laser radiation is positioned at a 60° incident angle and an ion funnel was implemented to transport ions. The low mass resolution of less than 300 was still sufficient to analyse most of the ions. In later applications, this instrument was applied to laser desorption post-ionisation for investigations of molecular species.

Using an orthogonal ion extraction method and a linear TOFMS, Hergenröder et al., with a home-built LIMS system, investigated the performance of both femtosecond and nanosecond lasers for laser ablation/ionisation [18]. The sample is placed vertically at the side of the entrance to the drift tube and the laser beam is focused on the sample at normal incidence. The instrument had a mass resolution (m/$\Delta$m) of 300 for $m/z = 65$. The instrument also showed that acceptable spectral quality can be achieved by applying low laser fluences. A few systems introduced here have demonstrated high instrument performance in delivering high-resolution mass spectra (2200, >10,000), highly sensitive measurements (sub-ppm), and successfully conducted chemical composition analysis. Their high performance was demonstrated in numerous performance and application studies relevant to geochemistry, material science, and bio-relevant applications [19,21,22,131]. We introduced two LIMS instruments designed in our group: a miniature LIMS instrument, (LMS) for space research, and a large laboratory LIMS system, LMS-GT [21,24,25,132]. Recently, another miniature LIMS system called ORIGIN equipped with ns-laser desorption source was developed for detection of amino acids on planetary surfaces [12]. All these instruments use a characteristic collinear or axial ion extraction method. The third instrument is high irradiance LIMS, LI-O-TOFMS equipped with gas collision cell and applying the orthogonal ion extraction method [20,23].

### 3.3.1. Laboratory: LI-O-TOFMS

A buffer-gas-assisted high-irradiance laser ionisation orthogonal time-of-flight mass spectrometer, LI-O-TOFMS, used in the Huang group, was successfully coupled with a femtosecond laser ablation ion source (Figure 4). The LI-O-TOFMS demonstrated high

sensitivity, high mass resolution and advanced quantitative capabilities for element analysis [20,129]. High laser irradiance (>$10^{10}$ W/cm$^2$) applied for the sample ablation produces high ion concentrations at reduced abundances of polyatomic species. Furthermore, the ion fractionation effects are reduced yielding improved quantitative chemical analysis of the samples. For conducting chemical imaging studies, the sample is remotely manipulated by an x,y,z-translation stage with micrometre positioning accuracy at the focal plane of the laser beam. Fast ions produced in ablation pass through an inert buffer gas cell (e.g., He at 800 Pa). This allows for reducing isobaric interferences due to multiple-charged species (charge exchange with the buffer gas) and polyatomic ions (delayed with respect to atomic ion bunch due to frictional forces with a buffer gas). However, only a small fraction of the ablated ions present in the plasma plume is extracted and transported via a lens system in the ion extraction region. Thus, laser ablation and ionisation in a low-pressure source with ambient gas and the subsequent orthogonal extraction and ion transport into the ion extraction region of the mass analyser can introduce additional fractionation processes. The ions are selected for the mass spectrometric analysis by pulse train data acquisition mode, applying a pulse width of 3 μs and pulse frequency of 40 kHz. This method allows one to select a front ion package and not delayed polyatomic ions for mass spectrometric analysis. Polyatomic ions arrive at the ion extraction region with a delay compared to atomic ions. The maximum kinetic energy distribution of polyatomic ions is reduced by the collision cell and only a small fraction of these species is mixed with the atomic ions selected by the extraction pulse. An fs-laser (S-Pulse HP, Amplitude System, France) with a wavelength of 1030 nm and pulse duration of 500 fs is employed in the laser ablation ion source. The femtosecond laser radiation was focused either by the lens or telescope system. A digital storage oscilloscope is employed with a recording length of 500 μs to match the flight time of ion packets of all the elements. With the orthogonal ion extraction from the laser ablation ion source, ion cooling combined with orthogonal ion extraction method, the mass resolution of 2200 was demonstrated [20,133].

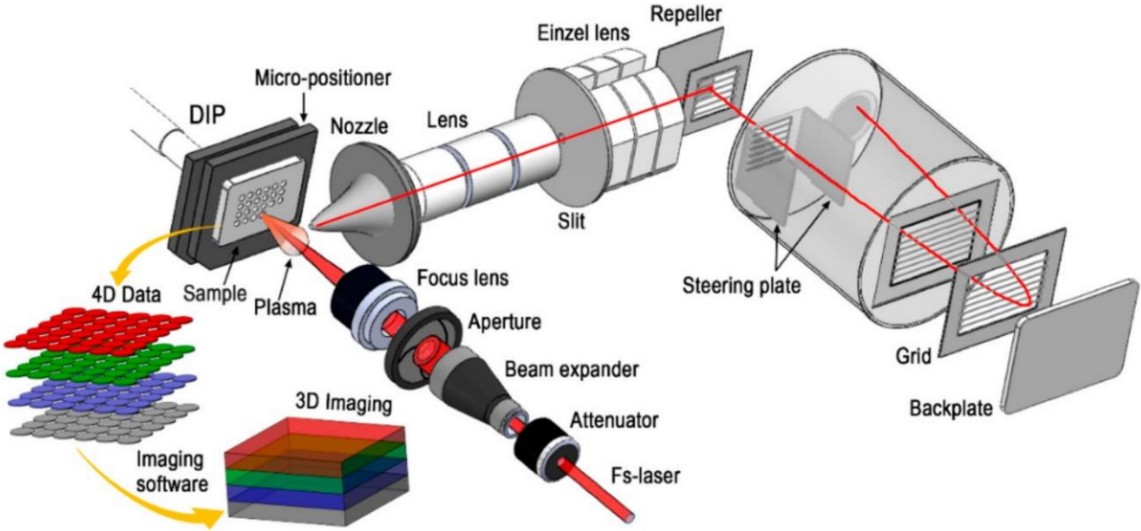

**Figure 4.** Schematic diagram of fs-LI-O-TOFMS applied in the Huang group. Reprinted with permission from [134].

### 3.3.2. Miniature Femtosecond-LIMS: LMS

The miniature fs-LIMS consists of a miniature reflectron-type time-of-flight (R-TOF) mass spectrometer kept in a vacuum chamber at pressure <$10^{-7}$ m bar and fs-laser ablation ion source based on Chirped Pulse Amplified (CPA) laser system (Clark-MXR Inc., Dexter, MI, USA) (Figure 5). Originally, this instrument was designed for the BepiColombo mission of ESA and since then it was used in the laboratory for a variety of investigations [21,25,132,135]. In the fs-LMS, fs-laser with pulses of ~190 fs pulse width at 1 kHz

repetition rate is applied. So far, studies with fundamental and harmonic outputs with 378 and 258 nm wavelength were conducted. The harmonic conversion was made using a commercial system (STORC, Clark-MXR Inc., Dexter, MI, USA). Furthermore, the laser optical system is modified to allow a double pulse ablation. The DP unit [88] consists of two beam splitters (50/50 transmission/reflection) used for the separation and recombination of the two laser pulses, as well as other optical elements, e.g., Al-coated mirrors to guide the pulses towards the sample. A dedicated fs-laser optical system consisting of dielectric and aluminium mirrors (e.g., Thorlabs Inc., Newton, NJ, USA) and beam expander (Eksma Optics, Vilnius, Lithuania) guides the laser pulses towards the chamber, through an entrance window, then through the mass analyser, and along its central axis, down to the sample, which is positioned right below the entrance of ion optics of the analyser. The laser beam is focused by a doublet lens and forms an ablation spot with a diameter of ~12 μm for IR at 775 nm, and ~10 μm for UV at 258 nm. The focal point of the incident beam is positioned about ~200–300 μm below the entrance of the mass analyser. The sample surface is remotely manipulated by an x,y,z-translation stage with micrometre positioning accuracy at the focal plane of the laser. The plasma plume is produced just at the entrance of the TOFMS allowing for a very efficient collection of ions into the analyser. The ion-optical system transmits only positively charged species towards the detection unit, which consists of two microchannel plates (MCPs) stacked in a chevron configuration and four centrosymmetric anode rings [136]. The signal is recorded by a high-speed ADC data acquisition system (U5303A, Acqiris SA, Geneva, Switzerland) with a sampling rate of 3.2 GSamples/s and a vertical resolution of 12 bit. The special design of the detector system, featuring a high dynamic range, allows signals from major to trace elements to be measured simultaneously [21]. Improved detection sensitivity of heavy trace elements could be achieved using a fast, high voltage pulse applied at controlled times to one of the electrodes of the ion optics system, to remove intense low-mass atomic ions. High-mass atomic ions in such arrangement can be sensitively measured by applying higher amplification on the detector system [137]. High-speed microstrip multi-anode multichannel plate detector system applied in LMS allows increasing the measurement dynamic range by merging the measurements collected from different anode rings [136]. The spectra can be recorded simultaneously but at different gains if necessary. With this method, the dynamic range can be improved to about 8 orders of magnitude [132,136,137].

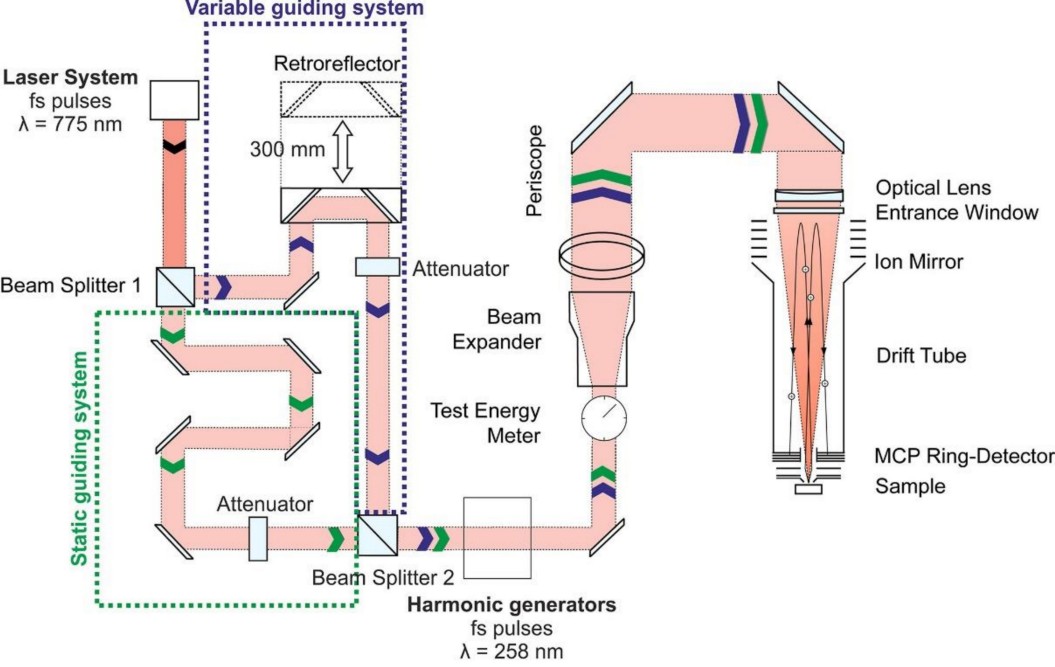

**Figure 5.** Schematic overview of laser ablation ionisation source with the mass analyser. Reprinted with permission from [82].

Further improvement of mass resolution and isotope accuracy was achieved using a post-processing filtering algorithm that uses systematic exclusion of individual distorted mass spectra from the data set to improve spectra quality [112]. Implementation of double pulse laser ablation ion source increased ion yield and also proved to be a powerful method in the suppression of isobaric interferences, improvement of depth profiling accuracy and quantitative performance to chemical composition [82,88].

### 3.3.3. High-Resolution fs-LIMS: LMS-GT

The high mass resolution fs-LIMS instrument called LMS-GT (GT stands for 'Gran Turismo') supports both superior mass resolution compared to other LIMS using TOFMS and ablation studies with the high lateral and vertical resolution (Figure 6) [24]. The instrument consists of an fs-laser ablation ionisation ion source capable of probing with spot sizes down to 1–2 μm, a long flight path folded twice by two ion mirrors and a channeltron detector.

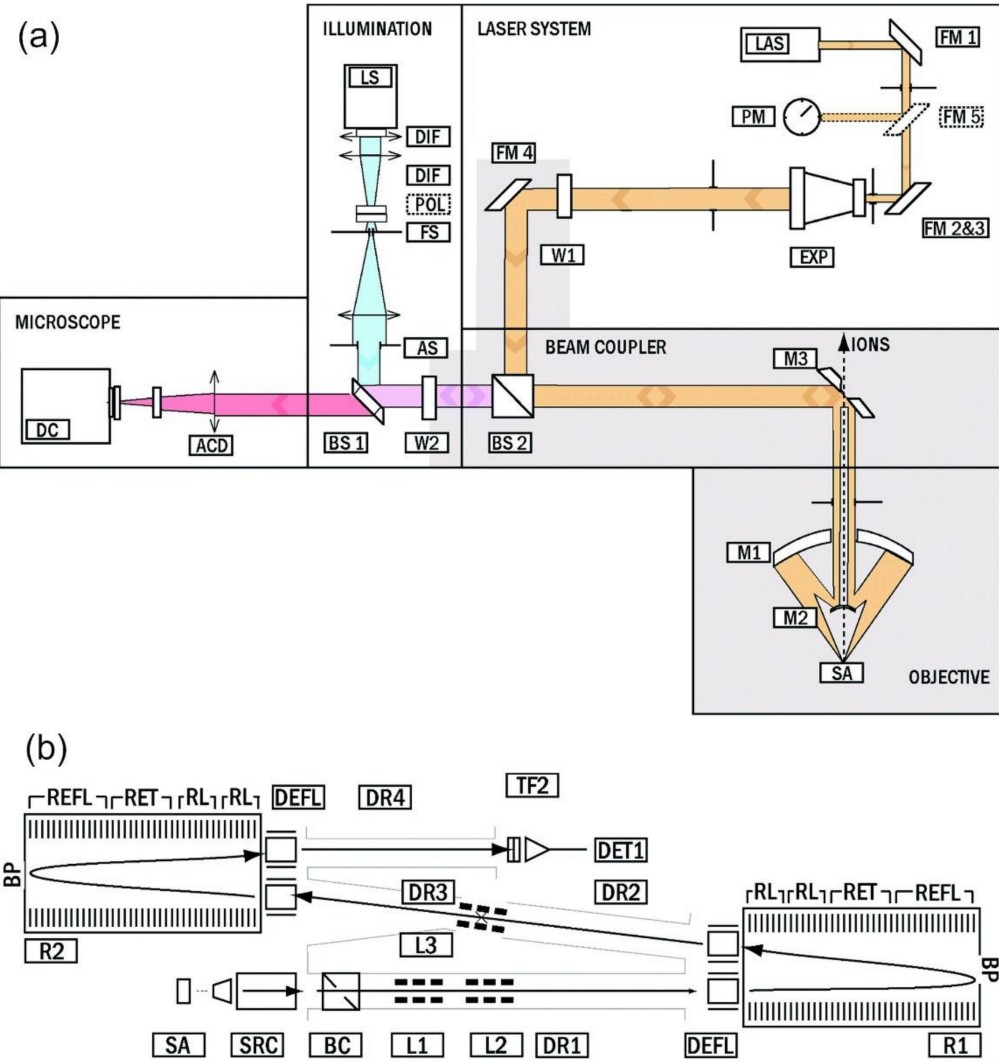

**Figure 6.** Schematic of optical and ion-optical system of LMS-GT; (**a**) The layout of the optical system applied in LMS-GT combines optics for laser ablation ion source, microscope system and the surface illumination system for the microscope observations. The Schwarzschild objective, supporting tight focusing conditions, is placed near the sample surface; (**b**) Ion optical design of LMS GT: SA: Sample location, SRC: ion source, DR1–DR4: Drift Tubes, L1–L3: Lens 1–3, DEFL: Deflection plates, R1&R2: Reflectrons, RL: Reflectron Lens RET: Retarder Section REF: Reflector Section BP: Backplane, DR: Drift Tubes, DET: Detector (for more details see reference [24]). Reprinted with permission from [24].

The optical system plays a central role in the instrument. It supports tight laser focusing on the sample surface and an in-line microscope system with a resolving power of 2 μm that allows verification of the ablation crater characteristics and location on the sample. This microscope can also be used to target specific sample micro-features during the measurement campaigns. The microscopy line uses the same objective as the laser. The instrument guides the laser radiation from the laser output to a two-mirror system known as Schwarzschild optics, where it is focused on the sample to a spot of 1–2 μm in diameter for applied IR fs-laser radiation. A Ti: Sapphire laser CPA system from Clark-MXR Inc. is used to produce femtosecond laser radiation (190 fs, λ = 775 nm). The pulse repetition rate is up to 1 kHz and the maximal output pulse energy is 1 mJ. The focusing of laser radiation is arranged by expanding the laser beam with a beam expander (Altechna 775 nm and 387 nm). The optical system design was optimised using ZEMAX tools [118].

To support optimal ion transport from the ablation spot into a mass analyser, the ion-optical design was merged with the light optical system. The ion-optical design of the instrument was entirely carried out using SIMION [138]. Custom-made MATLAB pre- and post-processors were made to create the initial ion populations and to visualise the simulation results. High mass resolution, in addition to high spectral quality, allows the removal of a significant fraction of isobaric interferences significantly improving the quantitative capabilities of the instrument. With the ion optical system, a large temporary spread of ion packet is effectively compensated with two ion mirror systems. Typically, the ion temporal spread at the ion detector in the LIMS system is a few ns close to the transit-time spread of the currently available ion detectors (<1 ns) [139,140] and the frequency bandwidth (~1 GHz) of the preamplifiers in data acquisition systems [132]. LMS-GT represents the system in which the ion packet temporary spread is in ns range, thus, comparable with these other time spreads. To reproduce the pulse shape of the signal produced by the detector, the sampling rate of the acquisition system should be 10 GSamples/s or larger. The LMS-GT instrument demonstrated mass resolving powers (m/Δm) of more than 10,000. With the mass accuracy in the 10 ppm range, the direct identification doublet or multiplet mass lines of isotopes, molecules, and clusters are possible. This removes isobaric interference effects, which are limiting the quantitative analysis of low mass resolution LIMS instruments. Trace element sensitivity is found to be in the ppm range. The optical system allowed efficient sample tracking during its mass-spectrometric surface investigation and micrometre resolution while conducting chemical imaging and depth profiling studies.

## 4. Performance of fs-LIMS Instruments in the Element and Isotope Analysis

Assessment of the capabilities and calibration of the LIMS instruments is commonly performed using standard research materials (SRM). NIST standards including high carbon steel, electrolytic iron steel, geological soils, and samples with certified isotope abundances are used. The following subsections cover different aspects of fs-LIMS performance evaluation.

### 4.1. Time of Flight, Mass Range, and Mass Resolution

The mass spectrum is measured on a linear time scale *t* by the ADC data acquisition system. For the miniature instruments, the TOF spectrum of elements from atomic mass 1 to about 235 is recorded within about 20 μs. These times are longer for large laboratory systems and can exceed 1 ms.

Figure 7 displays the mass range typically used in fs-LMS-GT and shows example of three high mass resolution mass spectra measured on three NIS standards.

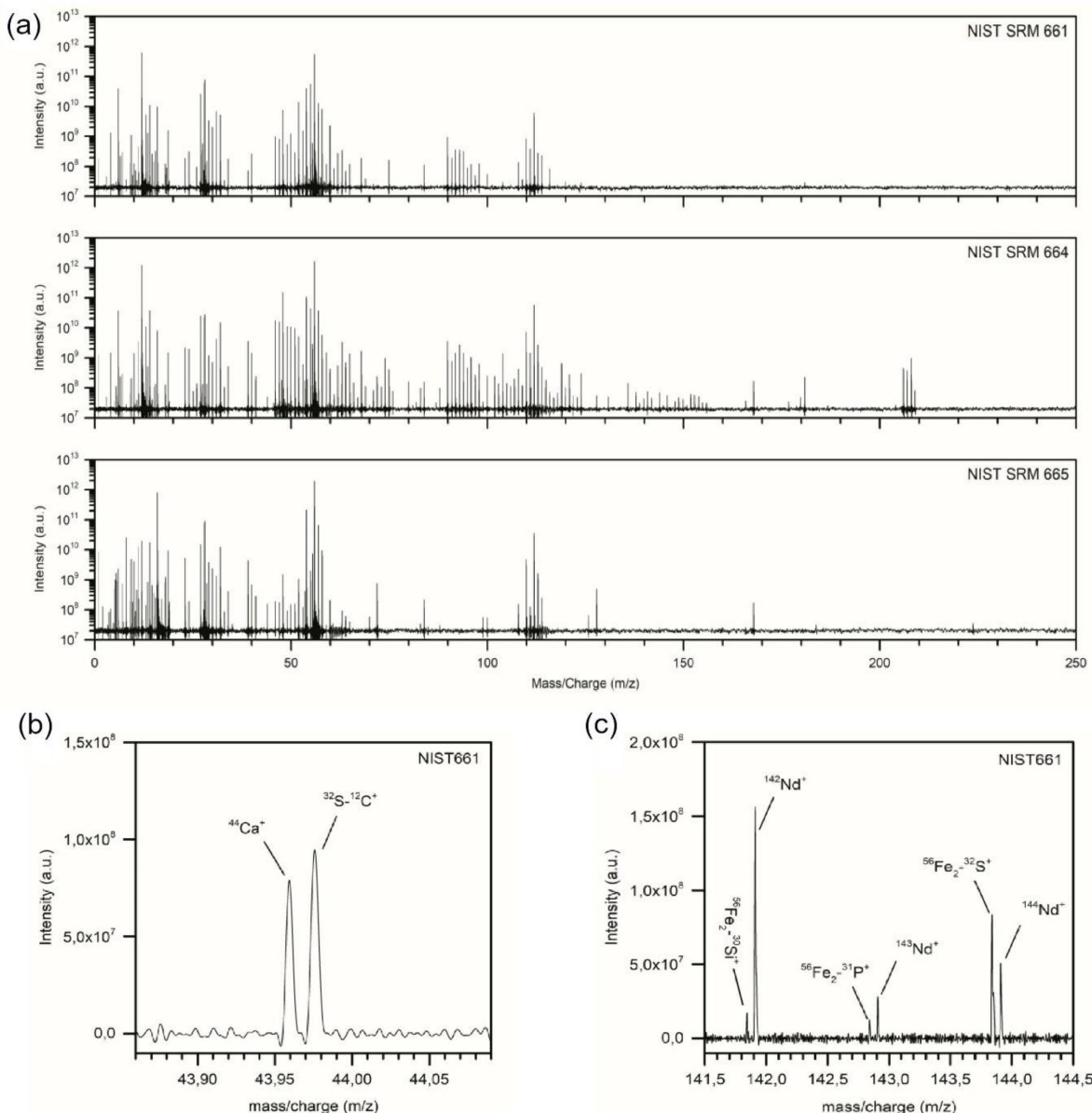

**Figure 7.** (**a**) Typical spectrum recorded with LMS-GT for three NIST SRM steel samples. The applied mass range is compatible with the applied data acquisition card to optimally use the sampling range of the card; (**b**) At the mass m/z = 44 both isotopic $^{44}$Ca and diatomic $^{32}$S-$^{12}$C mass peaks can be measured; (**c**) At the mass peaks 142, 143 and 144 in addition to isotopic Nd also diatomic mass peaks are observed [141].

The transformation into a mass scale is made with two calibration factors using an equation with quadratic dependence of mass *m* [atomic mass unit], on time of flight t[s] via $m = k_0(t - t_0)^2$. The coefficients $k_0$ and $t_0$ are the calibration constants. They can be calculated using parameters obtained from linear regression using the flight times for at least two isotope masses identified in the spectrum. They can also be determined from the ion optical parameters of the mass analyser. The mass scale accuracy (average absolute mass accuracy) can be calculated using the absolute values of individual mass $m_i$ errors and is given in part per million (ppm) [142]. This value for a miniature TOFMS is a few hundredths of ppm and lies in ppm or sub-ppm range for high mass resolution TOFMS.

In the earlier LIMS systems, the mass resolution (m/Δm) was limited, rarely exceeding 300 (LIMA 2A, LAMMA, LAMAS 10M, LAMS, LAZMA, MB TOF) [15,24,31,132,143–146]. Current high laser fluence LIMS designs have significantly improved mass resolution. For LMS, an instrument with about a 30 cm flight path, the mass resolution close to 700

in ablation mode [11,21,132] and 1100 in desorption mode [12] was obtained, whereas the measurements with LMS-GT can be conducted with the mass resolution of 15,000 and higher [24]. A new strategy to reduce temporal ion spread produced during laser ablation was applied in high irradiance LI-O-TOFMS instrument with inert gas cooling and orthogonal extraction [23,129]. While applying high irradiance laser ablation ion source, ion concentrations and ion kinetic energies are far too large to be analysed by the TOFMS directly. However, in this system, only part of the produced ions is analysed. The ions are extracted orthogonally and after cooling the selected ions in a low-pressure collision cell to temperatures of the cooling gas. Subsequently, the ions are transported by the lens system into the ion extraction region, where again a part of the incoming ions is used for mass spectrometric analysis. With this system, the mass spectrometric analysis is conducted with a mass resolution close to 2200 [23,129]. In miniature TOFMS systems, the mass resolution can be raised to $m/\Delta m \approx 800$ for the isotope ratio measurements by optimising the voltages applied to the ion optical system by specially developed "swarm particle algorithm" [132,147].

The dependence of the mass resolution on the laser irradiance is one of the most critical factors for miniature LIMS systems and is related to the space charge of the ablated plume (coulombic repulsion) and surface charging, both influencing the ion spreads [112]. In TOFMS systems, the mass resolution depends on the mass, but also other less controllable experimental effects, such as surface and space charging that can affect the mass resolution by introducing an ion packet temporal jitter due to varying fields at the surface and coulombic repulsion if the ion concentration becomes larger than critical density ($\sim 10^7/cm^3$), respectively. These effects are less severe in LIMS with a laser desorption ion source [135].

### 4.2. Dynamic Range, Detection Sensitivity, and Limit of Detection

The dynamic range within a single spectrum is defined as the ratio in the signal abundance of the largest and smallest useful mass peaks. The current LIMS systems report dynamic range based on summing (histogramming) multiple spectra together, which improves ion statistics, reduces background contamination and improves the measurement of smallest peaks. For the largest mass peaks, the acquisition system is important. Typically, ADC acquisition cards with an 8-bit or 10-bit vertical resolution at an 8 GSamples/s sampling rate are applied in the LIMS systems [132]. Without memory overflow, modern data acquisition firmware of the PCIe card supports a maximum summation of 64,000 spectra, which corresponds to a maximum vertical resolution of 24 bit and a maximal dynamic range of 7 decades per channel [132]. The firmware with extended memory offers a larger vertical resolution of up to 32 bit equivalent to a dynamic range of up to 9 orders of magnitude per channel. Practical limitations depend on the application. In the case of a miniature LMS, the summation of some tens of thousands of mass spectra shows a dynamic range of about 5–6 orders of magnitude by systematically averaging out the random noise. In the specific cases where a high dynamic range is required, the anode in the LMS detector system which consists of four separate active concentric segments can be used for a simultaneous readout of the signal [136]. Large laboratory systems typically have 6-7 decades of dynamic range [24,129].

Femtosecond LIMS can deliver sensitive measurement of chemical composition down to trace levels below ppm element abundance. Such measurements are of interest to many applications with a focus on either elements or isotopes. fs-LMS is designed to routinely achieve a detection limit at the ppm level and in some cases tens of ppb levels [132]. LI-O-TOFMS demonstrated limits of detection (LODs) in the $10^{-7}$–$10^{-8}$ mol/g range, corresponding to the sub-ppm level (for Li and Na, the lowest LODs were 0.2 and 0.1 µg/g which is about 30 ppb and 10 ppb atomic fraction for these elements, respectively). The measurements were conducted using the correlations between the known element concentrations and the measured signal intensities on artificial samples containing various elements at known concentrations. A dynamic range of $10^6$ in the measurements of various

high purity samples was achieved [129]. In another study from the same group, the element analysis of residues of mixed salt solutions were conducted [17]. The studies show that the instrument is capable of conducting trace analysis at the ppm level. For a single laser shot and an average ablated sample thickness of about 1 nm, the absolute limit of detection was determined $10^{-15}$ g (1 fg) for most metal elements.

*4.3. Chemometrics and Analysis Protocols*

For better understanding and control of the laser ablation process, the ionisation efficiency of elements can be represented as a function of several parameters characterising the chemical and physical properties of the element. Nonstoichiometric ionisation is caused mostly by matrix effects and these are directly related to the material properties [148]. The matrix effects were investigated in several different samples and the relative sensitivity coefficients (RSCs) of the elements in different matrices were determined [149]. By combining the RSCs values with the physical property values of the matrices, the datasets were analysed by the chemometrics tool of orthogonal partial least-squares (OPLSs) for ns-and fs-laser ablation. RSCs of the elements in different matrices were measured experimentally. A theoretical model was worked out by combining the laser–solid interaction and plasma expansion processes to predict the RSCs involved in the study. RSCs were derived based on Fe concentration as reference (see also Section 4.3 for definition). The theoretical expression for an $RSC_i$ for species *I* was developed with the parameters characterising the physical and chemical properties of the relevant element, including specific heat capacity, absorption coefficient, surface absorbance, fusion heat, vaporisation heat, boiling point, thermal conductivity and first ionisation potential; in plasma expansion. Moreover, the temperature and electron number density of the ablation plasma were calculated. The model predicts the influence of thermal properties on ionisation efficiency by ns-ablation. The non-thermal properties (e.g., ionisation potentials, IPs) contributed almost equally to the models for ns- and fs-laser ablation. This model was used to compare the predicted $RSC_i$ with that derived from the measurement results conducted on several geological standard research samples. Overall, a good agreement between theoretical and experimental RSCs was reported [150].

In our group, the data analysis protocol was developed specifically for the LIMS application supporting the automatic analysis of a large amount of data [151], typically in the context of 2D and 3D chemical imaging. The developed software allows fast and precise quantitative data analysis of time-of-flight mass spectrometric data. The quantitative analysis of isotopes, using automatic data analysis, yields results with an accuracy of isotope ratios up to 100 ppm for a signal-to-noise ratio (SNR) of $10^4$. The accuracy of isotope ratios was found to be inversely proportional to both SNR and mass resolution (m/$\Delta$m). Follow-up studies by our group identified that the individual spectral degradation can significantly affect the mass resolution and quantification while accumulating the spectra for improvement of the signal-to-noise ratio (SNR) in LIMS [112]. These are spectra not only suffering from accidental surface and space charging, but also detector saturation. Spectral degradation of the overall spectrum can occur if some individual mass spectra, affected by peak broadening, are included in the accumulation process. The studies present the method to automatically identify individual spectra subjected to peak broadening and a filtering method capable of systematic and reproducible exclusion of such spectra from the accumulation process. Improvements in the isotope accuracy of Si, Ni, and Cr by factors between 1.6 and 7.7 were demonstrated [112]. In very recent studies, the data analysis protocols based on the isotope intensity correlation and depth profiling data were developed in the same group. High-quality depth profiling data allowed a high-degree correlation between the isotope signals. The isotope ratio (δ-notation) can be derived from linear regression with accuracy and precision at the per mille level [152]. The data used for precise isotope measurements are improved using a spectrum cleaning procedure that ensures the removal of low-quality spectra. Furthermore, correlation of isotopes of an element is used to identify and reject the data points that, for example, do not belong to the species of interest or are affected by detection effects, such as detection

saturation, spectral distortion or insufficient S/N. The isotope intensity correlation method was successfully applied in LA-ICP-MS earlier, yielding an improved isotope accuracy and precision compared to more traditional methods [1,153].

A similar method of the atomic intensity correlation based on the depth profiling data was developed very recently for in situ calibrations of atomic abundances owing to insufficient atomic ionisation efficiency (matrix effects) [34]. The method can be applied to a highly heterogeneous sample where simple oxides or sulphides are present. The ablation of such molecular compounds, if sampling resolution is sufficient, results in simultaneous detection of atoms combined in the compound. These compounds can be present at various depths in a heterogeneous sample. Nevertheless, with depth profiling data, one can correlate atomic intensities and perform a linear correlation analysis. From the slope of this correlation line, the atomic abundance ratio is derived and can be compared with the expected ratio for a given compound allowing derivation of RSC factors for the constituting elements. With the intensity (isotope, atomic) correlation methods, the quantitative element and isotope analysis can be improved significantly, which is important for heterogeneous samples where the RSCs change with the local composition. Additional studies have to be conducted to understand the possible limitations of this method and if the accuracy and precision can be further improved.

### 4.4. Measurements of Elements and Isotopes

fs-LIMS equipped in TOFMS allows the measurements of all ions produced during laser ablation. Typically, the mass resolution and sensitivity are sufficiently high that most isotopes of a given element can be measured. The mass spectrometry, contrary to other analytical methods, displays a very simple structure, and mass peaks can be well-represented by a Gaussian peak shape. Both qualitative and quantitative information can be obtained from the measured mass spectra. Current fs-LIMS when applied at low or high laser fluences yield semi-quantitative results. To obtain quantitative results, corrections with RSC factors have to be applied. RSCs are derived either from measurements on matrix-matched standard reference materials as a reference or from in situ calibrations relying on correlation methods discussed above. Moreover, further improvements to the laser ablation ion source including optimisation of pulse duration, wavelength, laser fluence and laser irradiance will bring the RSCs toward the unit, which makes them obsolete [87]. The role of laser focusing on the quantitative performance has not been studied systematically yet. In a recent comprehensive review, various fractionation effects influencing quantitative analysis are discussed in detail [154].

Current LIMS instruments demonstrate capabilities to conduct measurement at a ppm level (part per million) and in special cases down to ppb levels (part per billion). The determination of isotope ratios differs from the element analysis in several ways. While concentrations of elements in a sample are the unknowns for element analysis, their prior knowledge is a requirement for precise and accurate isotope ratio determinations considering instrumental limits of detection. Further, the mass spectrometry should be conducted at a sufficient mass resolution and peak intensities to perform precise peak-integrated ion intensity determination. Finally, the correction for molecular and atomic isobaric interferences is critical to obtaining accurate isotope ratios if the mass analyser has an insufficient mass resolution. The isotope accuracy and precision in LIMS studies are typically stated in the range 1–5% [155] and are not sufficiently accurate in scientific applications. Except for measurements conducted by resonance-enhanced ionisation by (RIMS) [156–158], the studies by LIMS are also fragmentary [147,152]. fs-laser ablation has the potential to improve the accuracy of the isotope measurements. fs-laser ablation offers superior quality depth profiling data which can be used to perform isotope intensity correlations. The isotope ratio can be derived from the slope value of this correlation. This untraditional method used in LIMS data may improve the quantitative LIMS performance for achieving accurate isotope values [152]. Further studies on post-ionisation techniques

(fs UV double pulse) are needed as well to reduce matrix effects, isobaric interferences due to polyatomic ions and increase ion yields [87,88,152].

### 4.4.1. Measurements of Elements

The much weaker dependence of fs-laser ablation on physical and chemical properties of the material compared with ns-laser ablation allows for the investigation of an increased number of materials, as fractionation effects become significantly reduced [20]. This makes fs-LIMS less matrix-dependent and measurements of standard research materials can be avoided if the semi-quantitative information is sufficient. This is supported by results obtained with both, high and moderate fs-laser ablation by LI-O-TOFMS and LMS, LMS-GT instruments, respectively.

The LI-O-TOFMS instrument developed in the groups of Hang and Huang has been used in different types of semi-quantitative analysis of solids [129]. Low discrepancy RSCs for different elements were obtained for a series of artificial samples containing B, C, N, O, F, Si, P, S, Cl, As, Br, Se, I, and Te with He buffer gas pressure of 250 Pa and a laser irradiance of about $10^{10}$ W cm$^{-2}$.

In the miniature LMS, the accuracy of the quantitative determination of element abundances is verified against standard reference materials (SRM), typically by either analysing metallic samples such as NIST SRM 661, SRM 664, and SRM 665 [21,132,135], or analysing geological standards and natural samples [34,150]. To quantify the magnitude of fractionation, the relative sensitivity coefficient (RSC) for an element in the sample is introduced. The RSC for a given element is obtained by dividing the measured element concentration, $X_m$, by the expected concentration, $X$:

$$\mathbf{RSC_X} = \frac{\mathbf{X_m}}{\mathbf{X}} \qquad (2)$$

For well-defined chemical compositions, as is typically the case for NIST SRMs, atomic fractions are stated or can be derived from the provided atomic mass fractions. The mass spectrometric measurements yield atomic intensities and their fractions are calculated by dividing the atomic intensities of a given element with respect to all atomic intensities measured in the sample [135]. By comparing the measured fraction of element $X$ with the fraction of this element in the given sample, one obtains the RSC of the element. The observed dependence of ion production efficiency on elemental properties (e.g., first ionisation potential, electron affinity, mass number, melting temperature) in these experiments led to the development of chemometrics, which can be used to rationalise the observations [149].

In studies of Cu-Sn-Pb ternary alloys composition, fs-LIMS using an IR fs-laser radiation was employed. The abundances of elements determined from the mass spectrometric analysis were compared with the results obtained by inductively coupled plasma collision/reaction interface mass spectrometry (ICP-CRI-MS) and laser ablation inductively coupled plasma mass spectrometry (LA-ICP-MS) reference measurements. The two ICP-MS studies quote smaller errors but these error bars use aliquots of the sample (Table 3) [159].

Performance tests with the fs-LMS instrument investigating the accuracy of the quantitative determination of element abundances were conducted against reference samples [21,27,132,150]. Similar studies were repeated using double-pulse IR fs-LMS [87]. Figure 8 shows the correlations of the measured and certified abundances measured with (a): single pulse (SP) and (b): double pulse (DP), respectively. For truly stoichiometric results, all points would lie on the line with a slope of value 1. Deviation from this slope indicates nonstoichiometric atomic composition measurements which can be caused by various fractionation effects. From these data, the calibrations constants, the RSCs, for LIMS are derived.

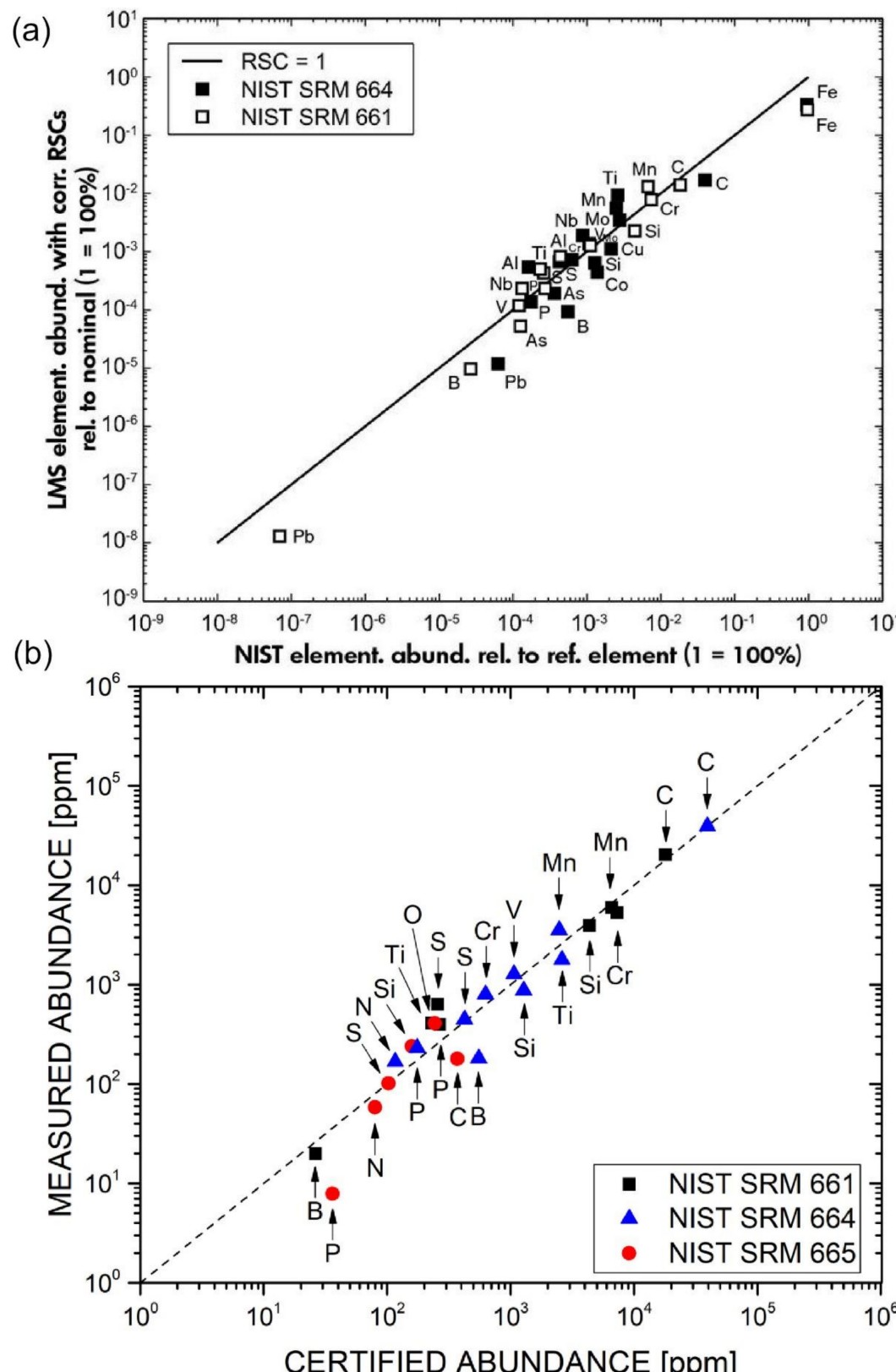

**Figure 8.** (**a**) Comparison of the measured and certified atomic abundances by SP IR fs-LMS; (**b**) Comparison of the measured and certified atomic abundances by DP IR fs-LMS. The Fe measurements are low due to the $^{56}$Fe signal saturation effects in the detector channel. Reprinted with permission from [21,87].

**Table 3.** Summary of composition measurements for three Cu-Sn-Pb alloy samples obtained by LMS, ICP-CRI-MS, and LA-ICP-MS techniques. Abundances and errors are given in wt% [159].

|  |  | DIN 1716 | LMS | ICP-MS | LA-ICP-MS |
|---|---|---|---|---|---|
| CuSn10Pb10 | Cu | 78.0–82.0 | 78.61 ± 11.87 | 82.26 ± 4.13 | 76.99 ± 0.75 |
|  | Sn | 9.0–11.0 | 9.94 ± 2.45 | 6.69 ± 0.79 | 10.91 ± 0.19 |
|  | Pb | 8.0–11.0 | 12.25 ± 3.28 | 11.13 ± 0.63 | 8.56 ± 0.47 |
| CuSn7Pb15 | Cu | 75.0–79.0 | 77.93 ± 11.77 | 75.55 ± 3.73 | 70.67 ± 1.03 |
|  | Sn | 7.0–9.0 | 5.65 ± 1.39 | 6.56 ± 0.40 | 7.30 ± 0.06 |
|  | Pb | 13.0–17.0 | 14.89 ± 3.99 | 19.45 ± 0.57 | 18.42 ± 1.04 |
| CuSn5Pb20 | Cu | 69.0–76.0 | 76.37 ± 11.53 | 75.63 ± 3.39 | 74.94 ± 0.68 |
|  | Sn | 4.0–6.0 | 5.99 ± 1.48 | 7.08 ± 0.73 | 6.92 ± 0.11 |
|  | Pb | 18.0–23.0 | 15.63 ± 4.19 | 20.59 ± 1.25 | 16.59 ± 0.62 |

Multiplied charged ions and polyatomic species causing isobaric interferences can limit the quantification capabilities of the LIMS system. A DP ablation ion source significantly reduces abundances of polyatomic species and increases atomic ion yield, thus improving stoichiometry of the generated plasma produced near the ablation threshold. RSCs for the investigated NIST steel alloys SRM 661, 664, and 665 are much closer to 1 compared to SP results, indicating the quantitative character of these measurements. Further improvement in getting the RSCs closer to 1 is expected using deep UV DP laser ablation ion source owing to lesser dependence of ablation rate on the material.

The influence of radiation wavelength on the quantitative performance was not explored systematically. Recent studies indicate, however, that matrix effects can be expected for special kinds of materials. The results from fs-LMS studies on silica chert performed with three different wavelengths show improvement of the ablation rate, ionisation efficiency, and quality of the ablated crater for UV fs-radiation. UV fs-laser ablation produces minimal heat-affected zones around the crater and material damage due to heat [160].

4.4.2. Measurements of Isotopes

Studies by the LI-O-TOFMS instrument conducted on metal and natural samples have demonstrated the isotope ratio measurements accuracies with less than 5–10 % deviation from the quoted values [161,162]. In other studies, the combined Thermal Ionisation Mass Spectrometry (TIMS) and LIMS studies on boron isotopes concluded the LIMS isotope accuracies similar to that derived from the TIMS method [163,164]. Some new inputs to the capabilities of LIMS to isotope measurements were provided by ns-LMS while investigating Pb isotopes in Galena samples [135,147]. With optimised laser ablation ion source conditions, the accuracy and precision at per mill level could be achieved as required for accurate dating. Similar results were achieved during measurements by TIMS. In recent studies conducted with IR fs-laser ablation ion source on lunar KREEP/zircon grains containing U, Th, and Pb, elements with concentrations in the range of 10s to 100s ppm, measurement quality was sufficient to determine the age similar to that measured by other techniques [165].

Early studies on NIST steel reference samples showed that the accuracy and precision of the isotope abundance measurements decrease with decreasing atomic abundance [21,132]. To measure relative isotope abundances with accuracies better than the per cent-level, the isotope abundances have to be larger than about 2 ppm, and at 100 ppb isotope abundances, a relative accuracy of 4% can be measured by optimising the ablation ion source parameters [132] (Figure 9A). Elements with concentrations above 100 ppm can be measured with an accuracy of per mill or better, as it has been shown for $^{206}Pb/^{207}Pb$ isotopes, whereas for isotopes with lower abundances values were in the per cent range [147].

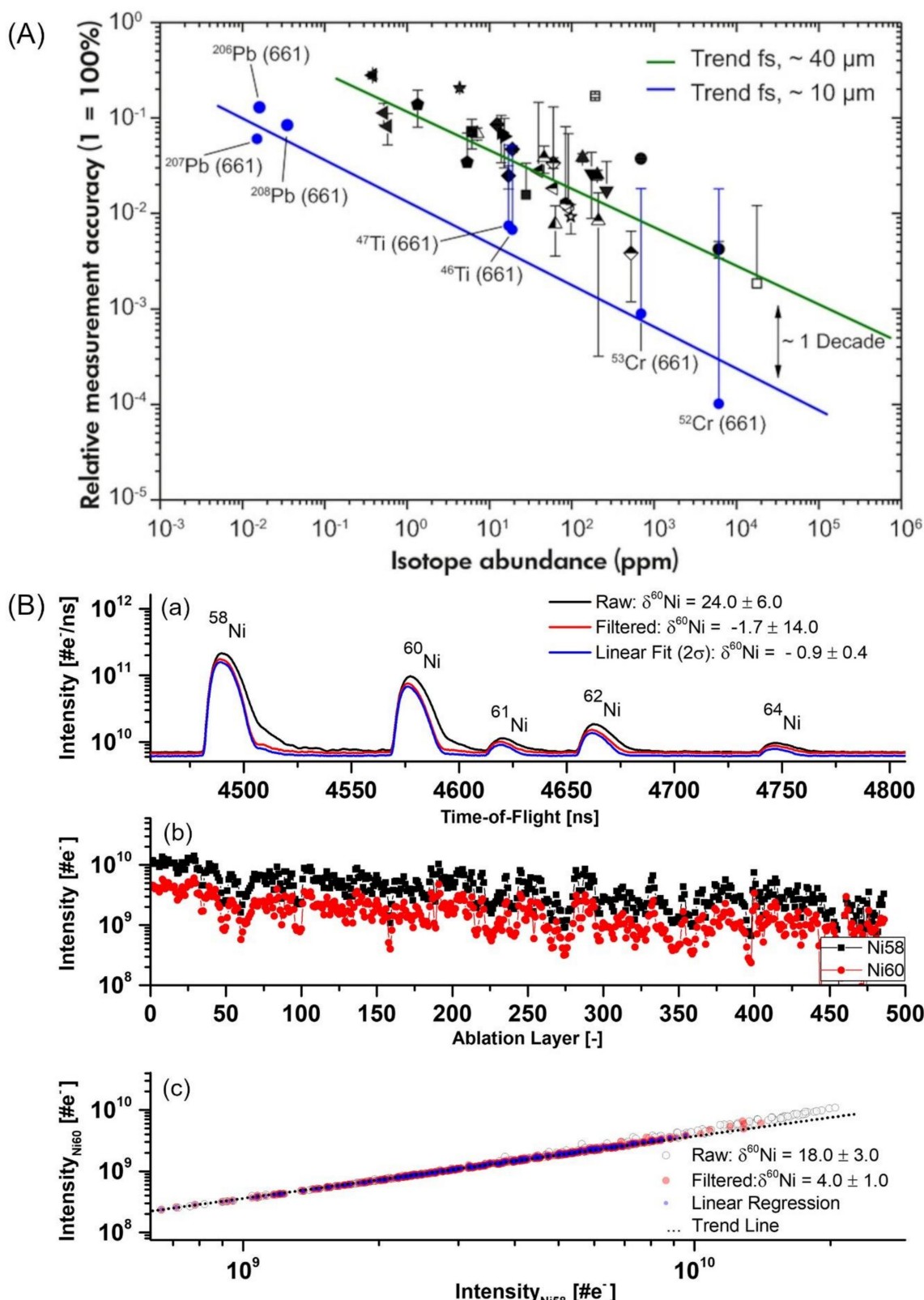

**Figure 9.** (**A**) Correlation between relative measurement accuracy of isotope ratios and isotope abundance determined from the mass spectra of the NIST SRM 661 steel sample measured by IR fs-LMS. Black and blue points indicate the measurements with laser ablation optics focusing to spot 40 μm and 10 μm, respectively; [27] (**B**) Isotope analysis of NIST SRM 658 Ni isotope standard sample showing the composite mass spectrum of Ni isotopes (**a**); depth profile built on $^{58}$Ni and $^{60}$Ni isotopes (**b**) and the intensity correlation curve (**c**). After filtering of the depth profile data, the accuracy and precision from 1% down to 1 per mill level can be achieved [152]. Reprinted with permission [27,152].

Recent UV fs-LMS studies identified several effects on the accuracy and precision of the isotope determination [34]. Compared to ion intensities produced in ns-laser ablation, the ion intensities produced by fs-ablation are significantly lower. When correlating the intensities for multiple isotopes of the same element, one should obtain well-defined linear correlations. Deviation from the linearity can be due to various effects, e.g., ion source instability, surface and space charge, detector saturation, and presence of isobaric interferences or local inhomogeneity of sampling area. A recently proposed data protocols allow the removal of spectra showing such effects and consequently improve quantification [112,152]. The studies conducted on micrometre-sized samples indicate that spectral filtering can be an important factor to achieve high accuracy isotope measurements [152]. Furthermore, determination of the isotope ratio from the slope using the linear regression method offers more accurate determination as from the mass peak integration alone [153]. A similar analysis method was applied for isotope ratio determination by LA-ICP-MS and improved accuracies were demonstrated in several studies [153,166]. The UV DP fs-ablation studies demonstrated (δ-notation) accuracies at the per mill level for measurements on an Ni isotope standard and several geological samples [152]. Similar accuracies were obtained for determining the Mg and B isotopes in the same samples. Figure 9B shows the method of analysis that allows for an improvement of the accuracy and precision of $\delta^{58}$ Ni-values by about a factor of 10, the values from the raw data with those obtained after the spectral filtering. The accuracy and precision of the measurements can be affected additionally by the presence of isobaric interferences due to the presence of polyatomic ions. The data show that they were significantly reduced while applying DP fs-laser ablation [112,152]. These initial results obtained by fs-UV DP LMS are promising for the application in the in situ geochemical research. LIMS can also be beneficial in the isotope analysis of micrometre-sized objects such as minerals or microorganisms, including microscopic fossils [152].

### 4.5. Depth Profiling, Chemical Imaging, and 3D Chemical Analysis

Due to the difference in laser-matter interaction, the required laser fluence for deepening ablation craters increases with increasing pulse width, i.e., from the femtosecond range to the nanosecond range. Additionally, the ablation crater quality improves with shorter pulse widths and lower laser fluences (close to ablation threshold) [44]. Thermal properties of the substrate play minor roles in the ablation of metals at femtosecond pulse width, mainly because heat diffusion into the metal is insignificant during the pulse's timescale. Compared with ns-lasers, the fs-lasers provide a material removal rate that is less dependent on the spot size, and the threshold fluence for ablation is both smaller and more precise for ultrashort pulsed lasers [167]. Table 4 shows laser ablation depth in several metals, including Ni, Cu, Mo, In, W, and Au obtained with 500 fs, 248 nm wavelength laser pulses at a fluence of 300 mJ/cm$^2$.

**Table 4.** Threshold laser fluence and depth per pulse for fluence 300 mJ/cm$^2$ derived from the laser (500 fs, 248-nm) ablation studies of several metals. Reprinted with permission [168].

| Metal | Threshold Fluence (mJ/cm$^2$) | Depth per Pulse (nm) |
|:-----:|:-----------------------------:|:--------------------:|
| Ni | 100 | 15 |
| In | 125 | 90 |
| Cu | 175 | 8 |
| Mo | 150 | 6 |
| Au | 210 | 17 |

Depth profiling data in mass spectrometry are obtained by collecting mass spectra from individual ablated layers. Hence, the sequence of mass spectra from a sampled spot is measured. For alloys or heterogeneous samples, the depth profile on the individual element is obtained by plotting signal intensity derived from each spectrum as a function of ablated layer number. Early fs-LIMS studies used depth profiling data to derive average depth profiling resolution. The sensitivity of LIMS to the analysis of layers as thin as a few

nm/pulse was demonstrated in studies in the early 2000s. Due to the low stability of their fs-laser ablation source, relatively poor precision of the analysis was achieved [169]. Multi-layered TiN–TiAlN samples were studied using fs-LIMS by Margetic et al. demonstrating sufficient resolution to resolve layers with thicknesses of 280 nm. In subsequent studies, Cr and Ni layers with thicknesses of 56 and 57 nm, respectively, [169,170] were measured as well. Systematic depth profiling analyses were conducted on metal foils and semiconductor wafers (Figure 10A). All expected metal ions appear in the mass spectra of moderate mass resolution of ~200. No attempt was made to calibrate the number of laser shots. An estimate of the depth resolution from a few nm to a few tens nm in the applied laser fluence range from 0.6 to 1 J/cm$^2$ [19].

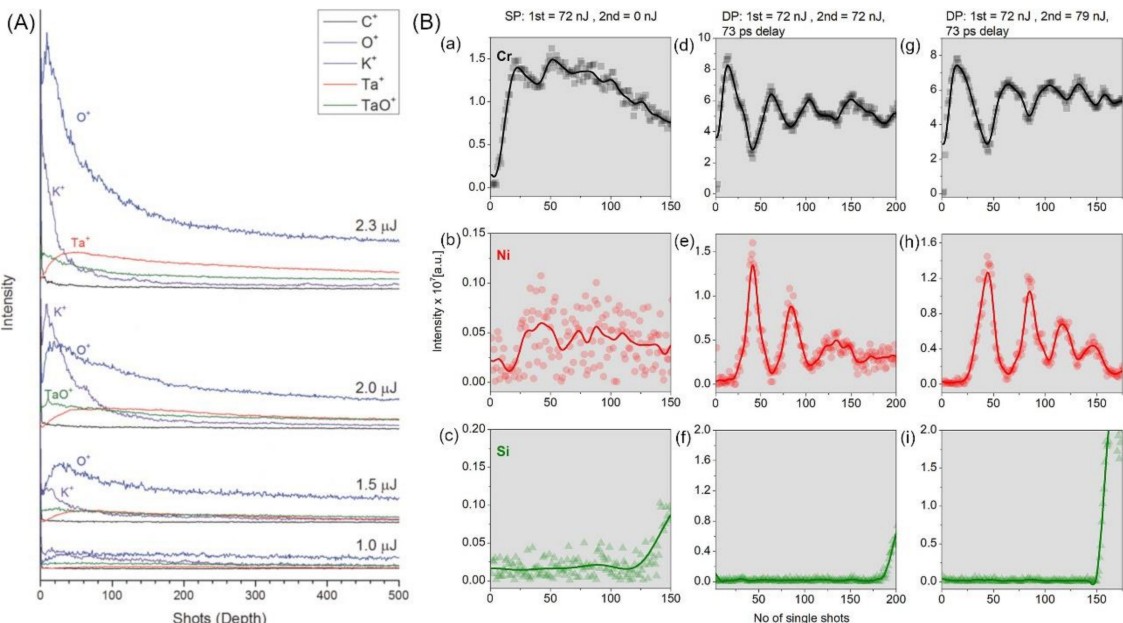

**Figure 10.** (**A**) Ion signal vs. laser shot number at various pulse energies. The data were obtained by ablating a ~330 nm Ta$_2$O$_5$ film deposited on Ta foil [19]. (**B**) Comparison of a single pulse (SP) (**a**–**c**), double pulse (DP) with both pulses at the same energy (72 nJ) and ~73 ps pulse delay (**d**–**f**), and DP with the first pulse at 72 nJ, the second pulse at 79 nJ, and ~73 ps pulse delay (**g**–**i**). Top, middle and bottom panels describe the recorded depth profiles of Cr, Ni and Si, respectively [82]. A clear improvement in the depth profiling resolution is seen between SP and DP operation. Reprinted with permission [19,82].

The high irradiance LI-O-TOFMS was used to investigate the depth profiling analysis of multilayer samples. Moreover, here, the superiority of the femtosecond laser mode for the depth resolution and trace element detection was observed. The fs-LI-O-TOFMS was capable of presenting the complete and explicit spectrum for each laser shot, performing depth profiling of coated layers with various thicknesses (tens of nanometres to tens of micrometres) examining samples with conductive and non-conductive samples. It was mentioned that a drawback of this technique was the need for sufficiently high laser irradiance for conducting both the ablation and ionisation [23].

Several experiments on the depth profiling performance were conducted with the miniature LMS. Single laser shot mass spectrometric measurements were conducted on electrochemically deposited Cu films of a defined thickness on blanket wafer coupons to investigate the correlation between laser pulse energies and resulting mean ablation rate [171,172]. These test beds were prepared under galvanostatic conditions in the presence of certain plating additives (e.g., polymers and surfactants), which is based on industrially applied protocols in the semiconductor industry for the fabrication of Cu interconnects in microprocessors. Some Cu films were prepared at slightly different plating conditions in which an embedment of the additives inside the Cu deposit is observed in form of spatially confined contamination layers with few nm thicknesses (further detailed discussion is

given in Section 5.3). The femtosecond laser system was used as an ionisation source with a spot diameter of about 15 µm. At higher laser pulse energies, an increase in the ablation rate was observed. The mean ablation rate showed a logarithmic dependence on the laser pulse energy, which however was not affected by the presence of impurities. In contrast to traditional SIMS measurements, where a certain area of the sample is typically scanned to monitor these impurities, in LMS, it was sufficient to probe single locations (size of the ablation crater) to identify and measure the embedded nm-thick impurity layers. Whereas the impurity intensities recorded by SIMS expanded over a broader depth range and attenuate with progressive ion erosion, LMS allowed for a smooth ablation procedure with negligible roughening effects at the crater bottom (Cu film thickness of about 15 µm). Surface roughening is provoked by the ion sputtering of the SIMS, which smears and dilutes the embedded contaminants in the subjacent Cu layers. The smoother ablation enabled the identification of a specific elemental pattern for the individual layers characterising the chemical composition of the contaminants at the location of their embedment. Another drawback of the commonly applied SIMS technique for this type of samples is the presence of strong matrix effect that makes the stoichiometric analysis of the embedded impurities impossible. Fundamental laser ablation studies were also carried out on Si, a dielectric material with different ablation behaviour, and were compared to Cu. To study the size and depth of craters created by the laser ablation process, an anisotropic etching method was applied to Si to cross-section the craters. Cu craters were analysed by applying a PDMS casting procedure which creates an imprint of the craters using specifically developed multi-component polymers [173]. Both techniques demonstrated the formation of cone-shaped craters with depths from 1 to 70 µm, depending on the number of laser pulses. The ablation depth was larger for copper and exceeded the ablation depth for Si by roughly a factor of 3. The high sensitivity of the instrument allows studies on the ablation threshold. At these conditions, the removal of 1 nm layer/laser shot can be conducted on average [173,174].

Recently, an improved depth profiling study was conducted with LMS by applying a double pulse ion source [88]. NIST SRM 2135c depth profiling standard reference material characterised by alternating layers of Cr and Ni (5 Cr layers, and 4 Ni layers) was used to conduct the depth profiling performance studies [82]. Comparison of single pulse (SP) and double pulse (DP) UV-laser (258 nm, pulse duration ~190 fs) ablation showed improved depth resolution for the latter irradiation method. This can be achieved by tuning the second pulse delay (~73 ps) and pulse energy (~72 nJ/pulse). The DP setup allowed for about 15- and 5-fold signal increase for Ni and Cr signals, respectively, compared to SP. All individual Cr and Ni layers could be resolved, and a mean depth resolutions of ~30 nm for Ni and ~37 nm for Cr could be achieved (Figure 10B) [88]. These studies show that fs-LIMS can readily compete with state-of-the-art LA-ICP-MS instruments [175].

Current x, y, z translational stages can position sample surface against the laser focal spot with desired precision and accuracies down to tens of nm. fs-laser ablation typically produces low surface contamination in near ablated crater area. With well-defined crater shapes, the next ablation spot can be placed near the beforehand produced crater without affecting quantification. Based on this observation, a 2D multi-position binning mode was proposed to sequentially probe the entire sample area of interest layer-by-layer [176]. Arrays of electrode-posited micrometre-sized Sn/Ag solder bumps served as a test bed for the method development. The presented studies show that the layer-by-layer approach outperforms the single crater analysis protocol when inhomogeneity is non-uniformly distributed along with the sample, and the probing ion source size has a similar dimension to the chemical components of interest. Increase of the lateral measurement spot statistics led to increased sensitivity with a signal enhancement of about ten times.

With a repetition rate of fs-laser systems and fast data acquisition systems, a rectangular area of 1x1mm$^2$ size can be analysed within several hours with crater sizes of about 10 µm. Typically, the time for moving the stage from one to other position limits is comparable with the measurement time at the one position [165]. The capability of fs-LIMS for chemical imaging which did not show considerable heat effects on the sample

(e.g., affecting zones and ablation rims) has been demonstrated in several studies [165,177]. Recent studies conducted on silica chert with fs-ablation applying IR (775 nm), near-UV (378 nm), and UV (258 nm) clearly demonstrated that UV fs-ablation is preferable for this material [160,178]. Figure 11 shows ablation spots resulting from these three different laser wavelengths observed by optical microscopy. The crater quality for UV fs-laser ablation is best and the measurement results in well-defined craters. Furthermore, an improved sensitivity of the deep UV fs laser ablation mass spectrometric measurements was demonstrated in these studies.

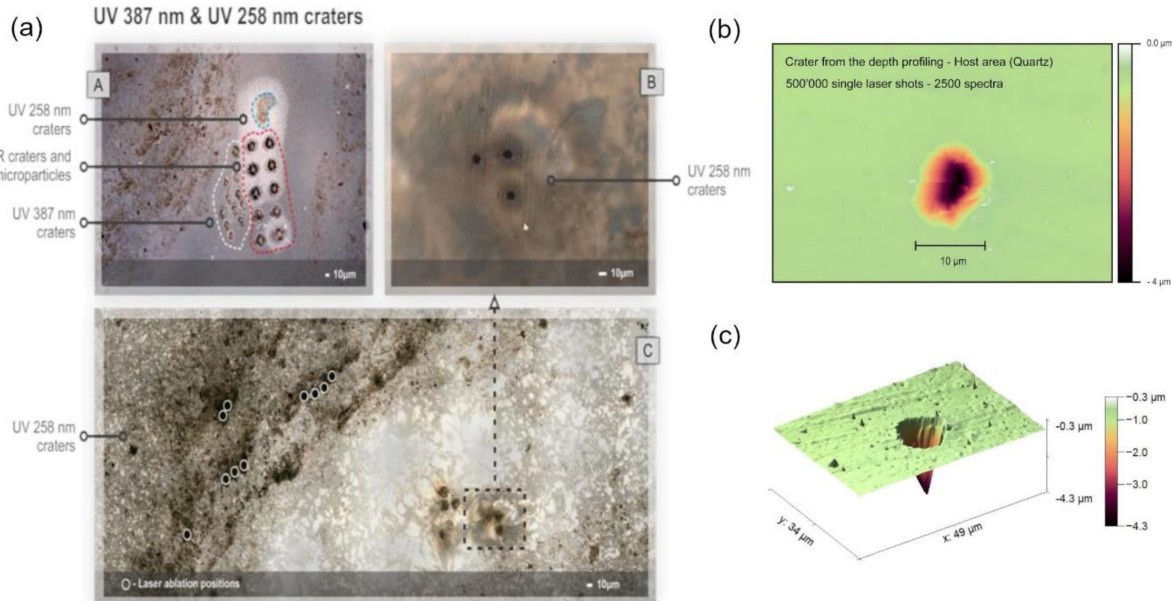

**Figure 11.** Panel (**a**): (**A**) Craters produced with different wavelengths. Red encircled area–IR 775 nm craters, white encircled area–UV 387 nm craters, and blue encircled area–UV 258 nm craters. (**B**) Close up look at the UV 258 nm craters. (**C**) Panoramic view of Gunflint chert sample with marked positions of UV 258 nm craters. Panel (**b**): Microscope z-scan image of the crater acquired from the depth profiling campaign with UV fs-laser radiation. Panel (**c**): Crater structure formed in this study [160,178].

Current fs-LIMS can conduct 3D chemical analysis by combining chemical imaging and depth profiling measurements [134,159]. High-resolution 3D chemical analysis down to the sub-micrometre level in all spatial dimensions seems to be possible, as well as taking into account the performance of the current fs-LIMS [24].

## 5. Applications

### 5.1. Geological and Geochemical Samples

LIMS represents one of the few unique instrumentations that do not require sample preparation before mass spectrometric measurements and offer a direct analysis of geological samples. LIMS can be used for bulk analysis, studies of subsurface layers, and measurements of the chemical composition of micrometre-sized mineralogical grains or thin layers. In geological applications, the knowledge of element and isotope redistribution within geological samples yields means to understand the geological processes contributing to the material formation (melting, solidifications, weathering and changes in climate). Rocks and soils contain the geological record of the processes that shaped the planet (planetary evolution) which can be deciphered through their chemical composition. Major and minor elements in igneous rocks are bound in a variety of oxides and are essential structural elements in most igneous minerals and planetary crusts. Some elements may be abundant in specific rock types being major elements (e.g., the carbon in coal and carbonate rocks). Determination of the chemical composition of geological samples down

at the ppm and ppb levels helps in understanding the physical and chemical evolution of the planet and origin of the solar system [116].

Usually, high inhomogeneity of the geological samples makes the quantitative geochemical analysis challenging. For reliable bulk analysis, geological samples are commonly homogenised and investigated with matrix-matched standard reference materials to obtain necessary RSCs. All techniques suffer from the various fractionation effects challenging [155] the element or isotope composition measurement. Improvements to stoichiometric atomic ionisation can be made applying e.g., double pulse ablation preferentially with UV fs-laser radiation. Instead of parallel measurements on standard research materials, one can also conduct in situ atomic abundance calibration, as discussed above. Recent data analysis protocols indicate that such corrections for matrix effects and other ion detection limitations can be conducted in situ even on highly heterogeneous materials. The selected studies discussed below demonstrate high capabilities of current fs-LIMS of investigating various geological samples.

Already in earlier studies using an ns-laser ablation ion source, LAMAS10 (Sysoev group) and LI-O-TOFMS (Huang group) demonstrated highly sensitive analyses on geological bulk and powdered standards after optimisation of several experimental parameters [179,180]. However, these studies also show that element fractionation can be high while determining RSCs. It has been shown that the majority of elements can be determined at a semi-quantitative level without using standard reference materials. Initial performance studies on miniature LMS with ns-laser (532 nm) ablation source did not only demonstrate high sensitivity of the instrument close to a few ppm (part per million in atomic fraction) but also indicated large fractionation effects while investigating a highly inhomogeneous meteoritic sample [135]. Similar results were presented in other miniature LIMS instruments, although obtained with other arrangements of the ion optical system of the TOF mass analyser [30,143]. The high irradiance LI-O-TOFMS with buffer gas demonstrated high performance to geological sample measurements by applying an ns-ablation ion source. Rapid detection of multiple elements in geochemical standard reference soils with little or no sample preparation was conducted with the semi-quantitative ability covering a range of 6 decades of dynamic detection. After obtaining the RSCs, the measured concentrations of all elements were closer to their certified values [181]. With a similar goal, other studies on geological samples were conducted by IR fs-LMS for fluences at about 1 J/cm$^2$ [150]. The four different reference samples were analysed and the consistency of the RSC factors could be demonstrated. IR fs-LMS proved to deliver semi-quantitative measurements and can be used for in situ quantitative chemical analysis of rocks and soils with appropriate calibration. In the early studies, ns-LMS instrument demonstrated the quantitative performance for isotope composition measurements [147]. By optimising the laser fluence for ablation on Galena mineral and the data acquisition procedure for isotope measurement, the accuracies at the per mil level and over were achieved in the measurement of Pb isotopes. Similar isotope compositions were measured using TIMS measurements conducted on the same sample material for reference. ns-LMS can also be considered as the instrument capable of accurate dating using the Pb-Pb method.

Several studies on geological samples were devised to understand the nature of small micrometre-sized inclusion and filamentous structures embedded in rocks [32,35]. The sample of micrometre-sized filamentous structures embedded in an aragonite crystal was investigated initially by IR fs-LMS. The sample was obtained from the floor of the Atlantic Ocean. The initial microscope inspection indicated the possibility that filamentous structures may be ancient microscopic fossils. The sample was investigated using the imaging capability of fs-LMS [35]. These studies identified the structure using its intrinsic chemical (element, isotope) composition different from that of the host mineral. Moreover, using the depth profiling method, further isolation of the chemical composition of the filament from the composition of host mineral was achieved. Although the presence of bio-relevant elements in the filamentous structures indicated the possibility of bio-origin of these structures, identification of possible isotopic composition and bio-relevant

fractionation would be necessary to support this identification. However, the improved accuracy and precisions would be required to prove this statement [35,88]. Subsequent UV DP fs-laser ablation LMS studies were conducted allowing improved ionisation efficiency and reduced isobaric interference by polyatomic species on the isotope compositions of B, Mg and Ni elements. Using the new data analysis protocol based on the intensity of isotope, correlations obtained from the depth profiling data, accuracy and precision of the isotope ratio (δ-notation) at per mille level was achieved. These accuracies allowed conclusions on ancient ocean conditions (B) and the classification of rocks (Mg) [152]. The measured Ni isotope ratio indicated bio-origin mechanism for isotope abundance change. The performance of fs-LMS to deliver quantitative chemical measurements of micrometre-sized objects is accounted to its high sensitivity and several improvements to the laser ablation ion source (UV irradiation, DP).

In two other studies with IR fs-LMS, a mineralogical inclusion in amygdale pillow basalt obtained from the ocean floor was investigated to understand the nature of the filamentous structure observed by an optical microscope. In the first study, fs-LMS measurements were accompanied by several measurements by other techniques, including Raman spectroscopy and environmental scanning electron microscope [32]. In this study optical microscopy, spectroscopic and mass spectrometric methods were applied and comprehensive information about the mineralogical and chemical composition of the sample was obtained. IR fs-LMS depth profiling analysis added information that the inclusion consists of a highly oxygenated material and through the depth profiling analysis confirmed the presence of a range of mineralogical phases as concluded from earlier Raman and microscopic studies [32]. In the next IR fs-LMS chemical imaging and depth profiling studies, more detailed analyses of depth profiling data were undertaken [35]. A new data analysis protocol was applied based on the atomic intensity correlation on the depth profiling data to derive relative sensitivity coefficient and obtain the quantitative atomic abundances. Further ternary diagrams of the obtained atom intensities showed correlations of such atomic intensities to derive the mineralogical composition of the inclusion, similar to that established analyses by the geological community [34]. Key element abundances for chlorite and feldspar mineral groups are identified in these ternary plots, which is typically used to identify the mineralogical context in the analysed sample with the minimum number of chemical species necessary to describe the composition (Figure 12).

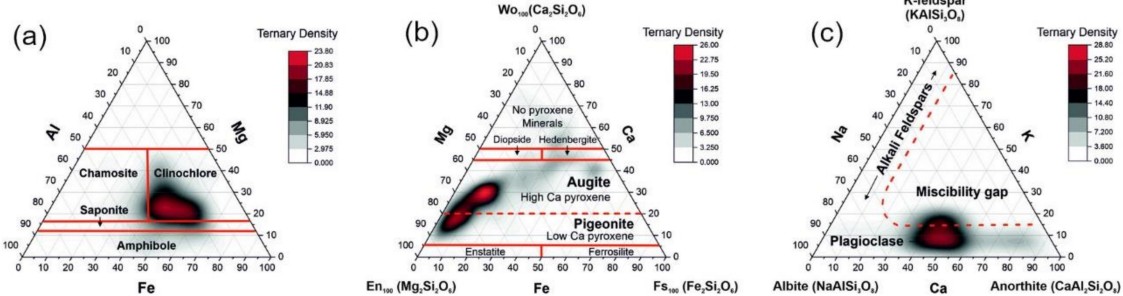

**Figure 12.** Ternary plots of the elements relevant to chlorite and feldspar mineral groups. (**a**) The correlation of Al, Mg, and Fe shows characteristic regions for various minerals of the chlorite group; (**b**) The correlation between Mg, Fe, and Ca abundance ratios indicates the presence of magnesium augite and pigeonite in the sample material; (**c**) The correlation diagram of Ca, K, and Na indicates the presence of the plagioclase feldspars in the inclusion material. Reprinted with permission from [34].

In other IR fs-LMS, studies accompanied by a high-resolution optical microscope setup were used to identify and characterise fossils of 1.9 billion years in a $SiO_2$ matrix of a Precambrian chert sample (1.88 Ga Gunflint Formation, Ontario, Canada) [28]. The use of a microscope significantly extended the performance of the LMS instrument, allowing much more precise MS imaging of a sample surface with a spatial resolving power of 1 μm and to acquire mass spectra with a spatial precision of about 2 μm. LMS also

successfully detected light elements and their compounds, such as $C_2^+$ and $C_xH_y^+$, which serve as indicators for the presence of fossilised microorganisms. Subsequent studies of Gunflint chert were undertaken to analyse chemical imaging data collected from two distinct zones—a silicified host area and a carbon-bearing microscopic fossil assemblage zone. A distinct chemical composition compared to the host mineralogy was measured with the identification of 24 elements. The study emphasises the necessity of applying the depth profiling measurement protocol for detecting the pristine composition of microfossils and identifying co-occurring fine chemistry (rare-earth elements) [160]. Next studies using UV fs-laser ablation ion source applied 2D chemical imaging of the Gunflint sample surface. The different areas of the host mineral and the microbial lamination surface can clearly be identified from their chemical composition. An advanced data analysis protocol of the chemical imaging and depth profiling data based on statistical correlation approaches such as weighted mass correlation network was used to isolate various fossilised forms [178].

In recent studies, the capabilities of fs-LMS to identify bio-organic material within geological samples were investigated in the context of instrument selection for the next planetary missions [33]. In total, 14 Martian mudstone analogue samples were investigated regarding their elemental composition. The instrument showed the capability to detect biogenic element signatures of the inoculated microbes by monitoring biologically relevant elements including, hydrogen, carbon, sulphur, iron. In correlation with other biologically relevant elements, the enrichment in carbon was considered as significant proof for the presence of microbes.

### 5.2. Meteoritic Samples

Cosmochemical analysis concerns the composition and evolution of matter. Planetary, cometary and asteroid material yield compositional details from which one can derive various stages of material transformation, underling chemical and physical processes since the time of the production of elements in star interiors [116]. In this respect, direct chemical analysis of extra-terrestrial samples, such as meteorites or samples returned from space mission, contributes much to this field. In such studies, the abundance of the major elements is typically important for mineralogical analysis, whereas minor and trace elements (isotopes) yield either an insight into the process leading to the formation of the material or are important for radiogenic dating. Again, high detection sensitivity at part per million levels or below is necessary for such studies.

Since many meteoritic samples are highly heterogeneous on small scales, spatially (lateral and vertical) resolved chemical analysis is most useful. fs-LIMS has the necessary capabilities to perform such studies concerning high detection sensitivity and high spatial resolution. Chemical analysis can be conducted on individual grains of mineral layers. Initial ns-LMS studies of Allende meteorite samples delivered spot-wise chemical composition, including the chemical mapping of the surface with closely spaced ablation spots [182]. ns-LMS mapping also allowed an insight into the mineralogical composition of Allende meteorite. Recent studies with IR fs-LMS expand former LMS studies and demonstrate a good correlation between the optical grain borders and the borderlines of element distributions, including the mineralogical context from a combination of elemental compositions in ternary correlation plots. Improved quality of the surface chemical mapping accompanied advances in determining the mineralogical bulk composition [183]. The abundances of C, O, S, Na, K, Li, Mn, P, Cr, Si, Fe, Mg, Ni, Co, V, Ca, Ti, Al and Sc were quantified in the bulk chemical composition of the matrix and its spatial chemical variability was determined. It was suggested that the obtained compositional data can be used to set constraints on models for the formation of the matrix material of carbonaceous chondrites in the solar nebula. A fast collapse of the dust-gas cloud following chondrules formation was inferred, similar to the mechanism for the formation of the parent body [183].

In other recently conducted IRfs-LIMS studies, a sample of lunar meteorite Sayh al Uhaymir 169 (SaU169) was analysed using a chemical mapping method. The measured geologically important elements were calibrated by relative sensitivity coefficients and

their abundance obtained in the studies were correlated with ternary graphs to determine mineralogy [165]. Additionally, a few heavier trace elements (Pb, Th, U) were identified in the mass on individual zircon grains which allowed for an age estimation [165]. Further, the estimation of the crystallisation temperature of the zircon grain was obtained from the Ti concentration distribution analysis.

IR fs-LI-O-TOFMS instrument was used for the spatially resolved multi-elemental imaging of a Nantan iron meteorite sample with 50 μm lateral and 7 μm depth resolution [134]. 3D scanning of the sample surface was accomplished by measuring 10 single laser shots spectra from one spatial spot of an individual sample layer. For calibration purposes, the measurements were initially performed using a self-prepared sample of high purity Cr, Fe, Ni, and Cu powders. Fe, Li, C, Na, Mg, Al, Si, P, S, K, Ca, Ti, V, Cr, Mn, Co, Ni, and Cu elements were identified in the mass spectrum and their abundances were determined without significant interference. The instrument was able to reach a detection limit of about $10^{-7}$ g/g and a dynamic range of $10^6$. Different spatial patterns for various types of elements in the meteorite reveal its preliminary composition, fractionation, and thermal history, as well as its surrounding physical−chemical environment during formation and evolution. Figure 13 displays elemental redistribution in the analysed sample volume.

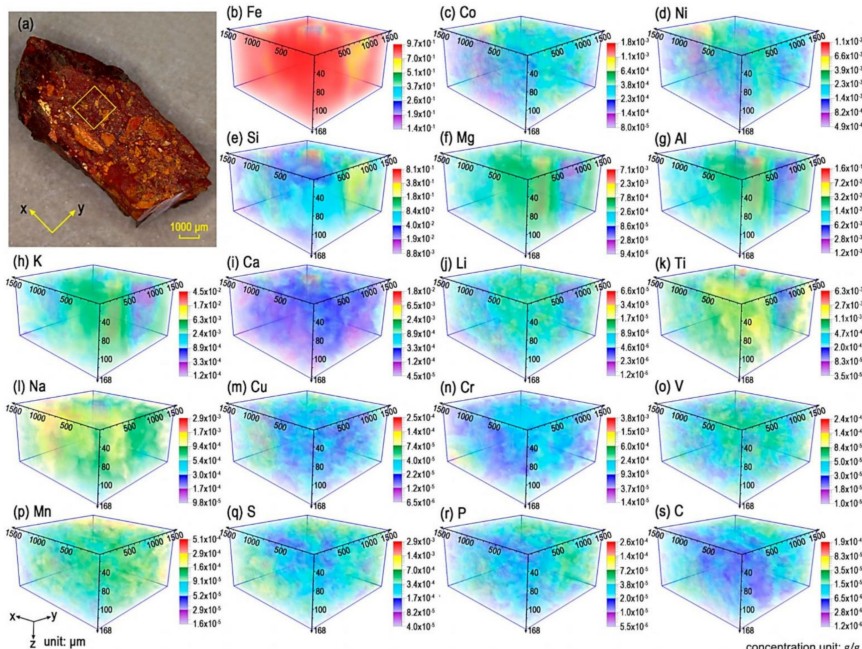

**Figure 13.** 3D element distributions recorded in the Nantan meteorite. The dimensions of the area are ~1.5 mm². (**a**) Photograph of the meteorite and (**b**–**s**) particular elemental distributions in the meteorite (g/g in concentration). Reprinted with permission from [134].

### *5.3. Material Science*

Many important processes such as corrosion, surface weathering, catalysis or surface contamination need information on the chemical composition of the material. Further, material purity control and contamination are an important task in preparation of metals and alloys, semiconductors, insulators (glasses, ceramics) and layered materials which are used frequently in the semiconductor industry. Again, due to its high sensitivity and multielement detection capabilities, fs-LIMS is an ideal method for such studies.

#### 5.3.1. Semiconductor Industry

Several studies with IR fs-LMS were conducted on model systems and state-of-the-art Cu interconnects important in the semiconductor industry. For the analysis of sample

purity, several analytical methodologies combined with depth profiling were developed. They are summarised in detail in a recent comprehensive review [184].

fs-LIMS was used to study an industrially applied process for the fabrication of interconnects in the microchip industry. To a large extent, the Cu interconnect technology relies on an electrochemical deposition process that involves the application of additives. This process is fast and cost-effective and generates a void-free bottom-up filling of the interconnects. On the other hand, a trend of these plating additives toward inclusion in minor and trace amounts into the deposit upon growth is known to be an important drawback that might impact the reliability, performance and lifespan of the end product. In this respect, it is of interest for the semiconductor industry to understand the reason for the embedment and provide new plating formulations with superior functionalities. A first attempt to solve this issue is to analyse how these additives are incorporated by measuring the chemical composition of the impurities inside the Cu deposits. Test samples consisting of 10 um thick Cu films were electrodeposited on a commercial Cu seeded Si (100) wafer in the presence of a prototypical two-component additive package consisting of a hybrid-type suppressor additive (Imep: polymerizate of epichlorohydrin and imidazole) and its essential co-additive (SPS: bis-(sodium-sulfopropyl)-disulfide). This additive package has the property to form within a certain current density range periodic setup and disruption cycles of a floating additive network on top of the growing deposit surface, which is observed to get incorporated into the Cu deposit each time the network is disrupted, resulting in a sequence of nm thick impurity layers along the Cu film. Such multi-layered Cu samples form an ideal testbed to optimize the IR fs-laser ablation ion source to conduct high-resolution chemical depth profiling methodologies with ablation rates at the sub-nanometre level per single laser shot [171,172].

In following IR fs-LMS studies, the incorporation of plating additives into Cu films was investigated by applying a depth profiling method to analyse the chemical composition and laser desorption technique to identify the molecular structure of the contaminants [185]. The embedded contaminants were observed to preferentially accumulate at grain boundaries inside the Cu deposit whereas the Cu grains remain largely contamination-free. Further, applying IR fs-LMS in the desorption mode, which requires about 10 times lower laser power, just above the desorption threshold, at the location of the impurity layers, the molecular structure of the contaminants was investigated. The results were consistent with the mechanism on the action of the hybrid additives interacting with thiolate-stabilized Cu(I) intermediates. In another study, the protocol for accurate spatially-resolved chemical analysis of actual state-of-the-art Cu electronic interconnects, so-called through-silicon-vial (TSV), was developed [186]. TSVs are large-scale Cu interconnects with high aspect-ratios (in this study 5 μm Ø × 50 μm depth) applied in the three-dimensional integration of multi-stacked transistors. Their particular dimension, geometry and multi-component nature (Cu channel surrounded by Si) make it difficult for quantitative analysis of the chemical composition. Depth profiling studies along the main axis of TSVs, as well as chemical composition analysis on the surface of cross-sectioned TSV structures, revealed a gradient in the carbon (C) content and indicated that the filling concept significantly deviates from common Damascene electroplating processes. The quantitative analysis revealed a C content that is similar to 1.5 times higher at the TSV top surface compared to its bottom. For a more comprehensive, understanding of the underlying mechanisms, further studies would be necessary [173].

The next study conducted by IR fs-LMS was done on electrochemically deposited Sn/Ag solder bumps, other important interconnects in the microchip industry [174]. Similar to the previously mentioned structures, the purity of the material, including the quantification of incorporated organic impurities, is of concern. Considerably improved analysis of this system was demonstrated by applying UV fs-ablation indicating that IR fs-ablation is less suitable for analysing low melting materials like Sn [81]. This study provided the first report on 3D-resolved chemical composition analysis on industrially high relevant Sn/Ag solder bumps. The 3D chemical imaging of the bumps was obtained by combining the 2D

binning approach along each layer of the entire bump (about 50 μm in diameter) and the depth profiling method in the depth direction (10 μm) (Figure 14). The quantification of the Ag content was found in very good agreement with ICP-MS studies, but in addition to the ICP-MS, which provides only bulk information, fs-LMS had provided micrometre level spatial resolution. In addition, the Ag heterogeneity, an undesired feature in solder bumps, was detected. The most relevant contaminants found in the solder bumps resulting from the plating process were O, C, and S, and their 3D distribution in the analysed volume was also determined [159]. Elaboration of matrix-matched RSC values for Sn and Ag from a certified reference material (BCS-CRM No. 347, BAS Ltd., Coulby Newham, UK) allowed for accurate quantification of the metallic phases of the alloy and also showed severe depletion of Ag on the surface. The studies contribute to the understanding of the SnAg plating process and the presented procedure can be important in additive performance screening.

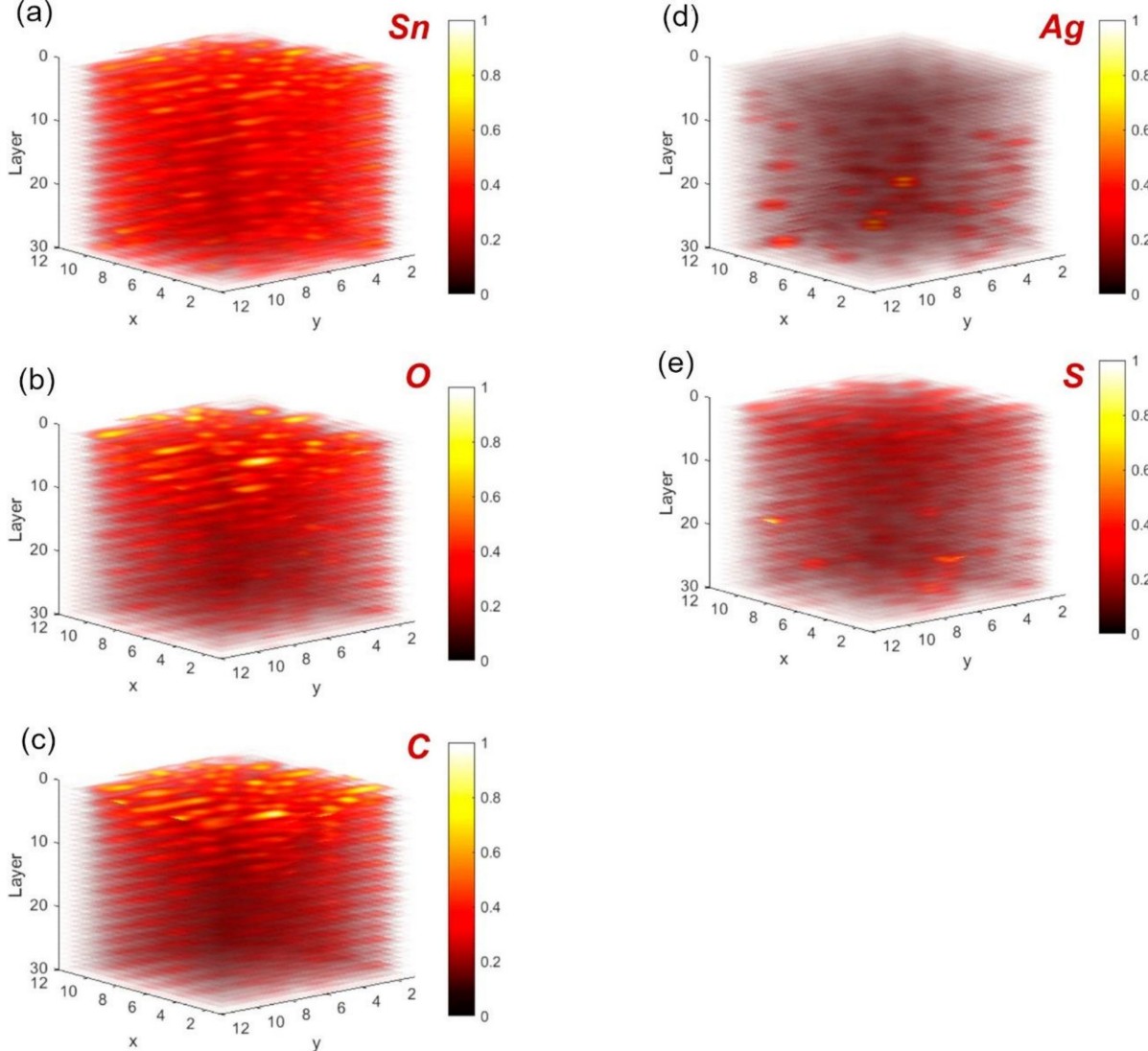

**Figure 14.** Signal intensity distribution of Sn, Ag, and the most relevant organic contaminants C, O, and S within the surface and early bulk region of the SnAg solder bump analysed by the 2D layer binning approach. Top row: depth profiles, bottom panels: 3D chemical imaging prepared on several elements. Differences in the alloy homogeneity, SnAg, and the spatial distribution of individual organic plating bath constituents (C, O, S) are apparent. Data have been normalized to the maximum signal intensity for this representation. Reprinted with permission from [81].

### 5.3.2. Materials for Electrochemistry

IR fs-LMS was applied to obtain the 3D chemical imaging of the Cu-Sn-Pb (lead bronzes: CuSn10Pb10, CuSn7Pb15, and CuSn5Pb20), model bronze alloys [159]. These materials were identified as high-performance cathode materials for electro-organic synthesis (dehalogenation, deoxygenation) of industrially relevant building blocks. The quantitative and spatially resolved element analysis of such cathode materials can be helpful in understanding the observed differences in the electrochemical reactivity and stability of these alloys. The element analysis showed specific composition correlations among the major elements (Cu, Sn, and Pb). On selected spots on the sample, the minor elements including Ni, Zn, Ag, and Sb, and trace elements C, P, Fe, and As were quantified. Inductively coupled plasma collision/reaction interface mass spectrometry (ICP-CRI-MS) and laser ablation inductively coupled plasma mass spectrometry (LA-ICP-MS) reference measurements were conducted in parallel. The element analyses showed significant chemical inhomogeneity in all three ternary bronze alloys with profound local deviations from their nominal bulk compositions and indicated further differences in the nature and origin of this composition inhomogeneity.

### 5.3.3. Ceramic Materials

The LI-O-TOFMS instrument equipped in VIS ns- and IR fs-laser ablation was used to investigate ancient ceramics [177,187]. Using ns-laser ablation ion source, a piece of Longquan celadon shard made in Song Dynasty and a piece of an imitation of ancient celadon shard were investigated using elemental composition analysis of the body and glaze form of both porcelain shards. The studies concluded that a difference exists in the element composition for different shards. Further, a piece of blue and white porcelain shard of Ming Dynasty was analysed by elemental imaging of Co, Mn, Fe, Ni, Ba, Ca, Mg, Na, Al, Si, P, K, Cu, Zn and Rb. The studies proved the capabilities of ns-LIMS for elemental imaging analysis and showed that this technique can be an important tool in porcelain provenance study and identification study of early precious porcelain. In the studies with IR fs-laser ablation source, both the spot-wise elemental and chemical imaging measurements were conducted on the porcelain body, as well as glaze of several samples originating from Yue kiln (in Southern China) and Yaozhou kiln (in Northern China) and different cultural eras. Basic sample preparation involved sherds slicing and cleaning with ethanol solution in an ultrasonic bath. The results indicate that Ti and Fe of the porcelain body and Ca, Fe, P, Mn, Mg of the glaze from Yue kiln can be considered as characteristic elements to classify porcelains among the different eras and discriminate them from the contemporary counterfeits. The high concentration of P also reveals that the glaze was fabricated out of plant materials. The differences in trace element concentrations of La, Ce, and Nd found in Yaozhou porcelain body from the Coin Ming Dynasty were interpreted as the historical record about the alteration of kiln site due to the warfare. The comparison of glazes between the two kilns revealed a difference in the aspects of the raw material, the structure of layers, and firing techniques, being responsible for the distinction of porcelain between southern and northern China. The elemental distribution in porcelain is usually inhomogeneous. The elemental mapping of a cross-sectioned Yaozhou porcelain helps to determine the elemental distribution in glaze transition body structure, which in turn helps in understanding the formation mechanism of the transition layer. Figure 15 shows the 2D-plots of elemental densities from the analysed cross-sections.

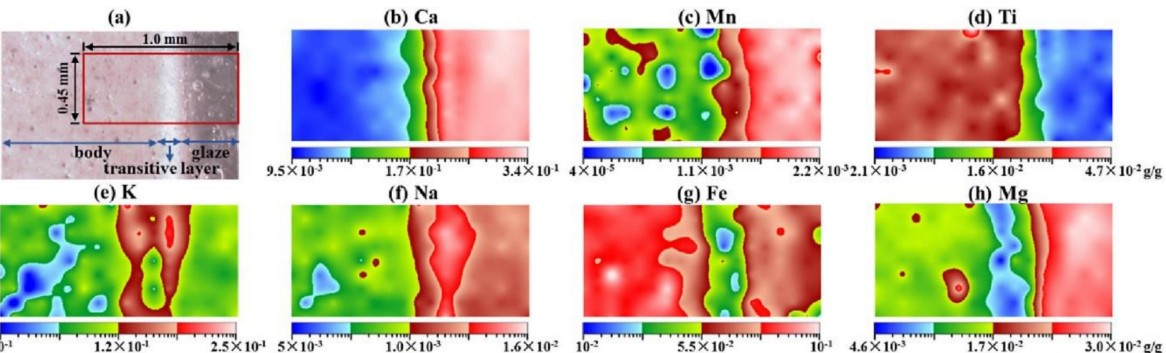

**Figure 15.** (**a**) Photograph of the cross-section of Yaozhou celadon. The elemental imaging in the area (1.0 mm × 0.45 mm) marked by red lines for (**b**) Ca, (**c**) Mn, (**d**) Ti, (**e**) K,(**f**) Na, (**g**) Fe, and (**h**) Mg. (For interpretation of the references to colour in this figure legend, the reader is referred to the web version of the article.) Reprinted with permission from [187].

### 5.4. Bioanalytics

Semi-quantitative element analyses of biological samples of tea leaf standard, Laminaria japonica, and pigskin were demonstrated in earlier studies conducted using the ns-LI-O-TOFMS instrument [188]. The detection limit of µg/g levels could be achieved for elemental determination with a solving power of about 5000. Low sensitivities for light bio-elements, such as C, N, H, and O, were observed due to the mass discrimination of cooling cell. Nevertheless, the signals of K and Ca were observed to saturate due to the high concentrations of these elements. Further, in addition to metallic elements, P and S were measured. Trace elements Sr and Ba were detected using their isotopes: [86]Sr and [135,136,137] Ba. However, the RSCs of most other elements were acceptable and the studies can be considered as semi-quantitative. The samples required a simple preparation; the pulverised tea leaf was dried and then pressed to produce a disk, the Laminaria japonica and pigskin were washed by deionised water and dried in a vacuum oven before the measurements.

In the recent studies, the LI-O-TOFMS system equipped with IR fs-laser ablation ion source was used to study the elemental composition of about 100 µm long, 40 µm in width, and few µm thick single Paramecium cells [189]. ICP-MS studies were undertaken to reference the LIMS results. The mass spectra were measured with the resolving power of 2200, hence little isobaric interference occurred due to clusters. Limits of detections were determined at the femtogram level. Before measurements, Paramecium cells were incubated in the solution for 24 h at 25 °C. Subsequently, living cells of paramecium were separated from large particles and debris by filtration using two mesh nylon sifters (pore size, approximately 30 and 150 µm) and gently and quickly washed with ultrapure water. The isolated cells were transferred and concentrated from the mesh onto a clean quartz plate. Replicate analyses showed that signal variations are 15–35% for metallic elements and 25–50% for non-metallic elements. The studies demonstrated that the instrument can conduct fast cell chemical analysis at the semi-quantitative performance.

## 6. Summary and Outlook

We reviewed the current progress in the development of fs-LIMS. Both improvements in the design of the mass analyser and application of fs-laser ablation ion sources contributed considerably to current LIMS performance. The strong dependence of ablation on the wavelength that is observed in ns-laser studies is reduced when using femtosecond lasers. Further improvements are achieved by applying UV (rather than IR) fs-laser radiation for the ablation. Laser ablation and ionisation under extreme-ultraviolet radiation (XUV) conditions were demonstrated as well in recent studies [190]. To date, there are no systematic wavelength dependence fs-laser ablation studies. The preliminary measurements suggest that fs-laser radiation applied at short wavelengths can further reduce matrix effects irrespective of their optical properties producing smaller and better-confined

craters, especially important for low-melting materials. The ion yield is higher by applying the fs-laser ablation compared to ns-laser ablation. The conclusions from these earlier fs-LIMS studies are consistent with the results of LIBS and LA ICP-MS that fs-laser ablation contributed to better performance figures of these techniques [38,39]. Improvements to stoichiometric ion production imply that chemical fractionation effects are reduced as well. No need for matrix-matched calibration was the conclusion from the studies of standard reference materials if the semi-quantitative chemical analysis is considered [115]. This is advantageous in studies of natural chemically complex and heterogeneous materials. Modified fs-laser ablation ion sources, either applied as flat-top, double pulse or modified by pulse shaping technology, have the potential of improving the control over the ablation mechanism by adjusting the laser pulse energy. The current studies show that the formation efficiency of polyatomic species can be reduced by adjusting pulse energy and inter-pulse delay.

The spatial resolution in chemical imaging and depth profiling analysis is observed to improve. For example, a tip-enhanced ablation fs-LIMS demonstrated 50 nm lateral resolution and effectively made LIMS nanoprobe technique such as nano-SIMS [119]. Similar spatial resolution is expected XUV radiation for ablation and ionisation [190]. The ease of controlling the laser spot size and thickness of the ablated surface combined with high ionisation efficiency provided by the fs-lasers allows for sensitive, spatially resolved, and 3D chemical analyses [134,159,178]. Furthermore, these data offer a new possibility for the information extraction by applying specially developed data analysis protocols, such as presented here, atomic or isotope intensity correlation using depth profiling data.

The continuous technological improvement in high-speed electronics and the development of fast data acquisition systems allowed considerable improvement of data acquisition limits and allows recording a large amount of information produced by the TOFMS. It is important to note that progress in big data analytics and unsupervised machine learning makes the generation of insights from large spectral databanks (3D chemical cubes and spectral libraries) increasingly easier and more accessible. One can also optimise the operation of a mass analyser to achieve optimal measurement parameters such as ion intensity and mass resolution. The control over the ion optical system after modelling optimised voltage settings can be further tuned experimentally to its best performance [132]. Fast measurements are important for chemical mapping, depth profiling analysis and, finally, 3D imaging. Developments in data analysis protocols show new possibilities of deriving important information reliably. This also indicates that LIMS potential of multi-element detection can be further explored to obtain new information. This opens new frontiers for improving the quantitative analysis and can pave the way to determining from the chemical analysis of complex mineralogical composition. Ongoing improvements in data reduction and compression strategies along with the development of more sophisticated and rapid hardware have greatly improved the TOFMS performance and also contributed to the current LIMS progress.

Development of LIMS is still progressing at a fast pace. There is potential in miniaturisation of the system with the appearance of miniature fibre-based fs-lasers and beam delivery technology based on fibre technology [99,191,192]. Then this system can truly be miniaturised and fieldable. The inconvenience of using a vacuum system exists but the requirements can be reduced to small volumes manageable by a small-sized pumping system. Miniature fs-LIMS proved its high-performance capabilities in various applications, including particularly chemical analysis of geological and meteoritic samples. In situ measurements on planetary surfaces would deliver valuable information with the qualities comparable to currently leading analytical instrumentation. LIMS is going to reach for the stars, both in space and on Earth.

**Author Contributions:** Conceptualization, M.T.; writing—original draft preparation, M.T.; writing—review and editing, all authors. All authors have read and agreed to the published version of the manuscript.

**Funding:** This research was funded by SNSF grant number [184657] and N.F.W. Ligterink was funded by Ambizione Fellow grant.

**Institutional Review Board Statement:** Not applicable.

**Informed Consent Statement:** Not applicable.

**Data Availability Statement:** Not applicable.

**Acknowledgments:** P.W. acknowledges SNSF grant number 184657; N.F.W. Ligterink acknowledges SNSF Ambizione Fellow grant.

**Conflicts of Interest:** The authors declare no conflict of interest.

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
