# Peer review of "Current Progress in Femtosecond Laser Ablation/Ionisation Time-of-Flight Mass Spectrometry"

_applsci, doi:10.3390/app11062562_

Round 1

Reviewer 1 Report

Tulej et al. reviewed comprehensive information on background, development, and various application of MS using femtosecond lasers. Extensive information is expected to provide useful information for understanding the advantages of using femtosecond lasers in the TOF-MS field, as well as the contents of the prospects to greatly contribute to related fields. I recommend publication of this review paper after revision of the minor parts below.

  • line 90: "Thanks to the efforts of a few groups, the development of LIMS did not stop after a hibernation period in the late 1990s.". I suggest removing "Thanks to the efforts of a few groups" for readability.
  • lines 139-150: Authors should add some references.
  • line 193:" ... become considerable High thermal energy electrons..." should be " ... become considerable high thermal energy electrons..."
  • line 231: "[49][38]" reference notation should be corrected.
  • line 235-236: "Figure 1 compares schematically different phases in laser ablation by ns- and fs-lasers." In figure 1 there is no ns- and fs-laser comparison.
  • line 260: "...intensity across the ablation area[44]." Add space between word and reference
  • line 269: "Early studies by Hergenröder et al. (2006) [19] compared ..." Author should exclude the year notation from the sentence.
  • line 272: "...gold and copper samples by Amoruso et al. (2002) [75], although..." Author should exclude the year notation from the sentence.
  • line 273-274: "ratios obtained by Hergenröder et al. (2006) [19]." Author should exclude the year notation from the sentence.
  • line 290: “(see Fig. 2 )” should be “(Fig. 2)”
  • line 298: “(see Table 1)” should be “ (Table 1)”
  • line 303: “(see also Fig.2)” should be “(Fig. 2)”
  • line 310-311: “by the Huang group (Zhang et al., 2013)” should be excluded.
  • Line 410-453: Authors should add some references.
  • Line 463:” … and ionization efficiency[87].” Add space between word and reference
  • Line 467:” … in material science[99].” Add space between word and reference.
  • Line 488: “… this spectroscopic method[39].” Add space between word and reference.
  • Line 493: “…over the plasma parameters[99,100].” Add space between word and reference.
  • Line 506: “…in recently reported studies[105].”Add space between word and reference.
  • Line 513: “…intensity tuning) are included[85].”Add space between word and reference.
  • Line 531: “self-focusing threshold (see also discussion in [44]).” Should be “self-focusing threshold [44]. ”
  • Line 535: “…(KTiOPO4)[106,107]. In” Add space between word and reference.
  • Line 541-548: Authors should add some references.
  • Line 558-574: Authors should add some references.
  • Line 578-588: Authors should add some references.
  • Line 621: “… during beam transport (see also discussion in [44]).” should be “… during beam transport [44]. ”
  • Line 636: “… spectrometric technique (see also discussion in [114]).” should be “… spectrometric technique [114]. ”
  • Line 756: “(see Fig.4)” should be “(Fig. 4)”
  • Line 799: “The DP unit[85] consists of two…” Add space between word and reference.
  • Line 816: “… centrosymmetric anode rings[129].” Add space between word and reference.
  • Line 820: “… be measured simultaneously[22].” Add space between word and reference.
  • Line 832: “… set to improve spectra quality[131].” Add space between word and reference.
  • Line 960: “… of ppb levels[125].” Add space between word and reference.
  • Line 1124: “aliquots of the sample (see Table 3) [155].” Should be “aliquots of the sample (Table 3) [155].”
  • Line 1175: “… abundance[22,125].” Add space between word and reference.
  • Line 1222: “… lasers[164]”Add space between word and reference.
  • Line1240: “…56 and 57 nm, respectively, [167],[166]” reference notation should be corrected.
  • Line 1242: “(see also Fig. 10a)” should be “(Fig. 10a)”
  • Line 1298: “…double pulse ion source[85].) ”Add space between word and reference.
  • Line 1383; “… meteoritic sample[128].”Add space between word and reference.
  • Line 1410: “… imaging capability of fs-LMS[36].”
  • Line 1441 “… microscopic studies[33].”
  • Line 1452: “(Error! Reference source not found.)” deleted this sentence and add the reference.
  • Line 1505: “… ablation spots[180]. Ns-LMS mapping” should be “… ablation spots [180]. ns-LMS mapping” Add space between word and reference.
  • Line 1512: “… composition[181].” Add space between word and reference.
  • Line 1595: “… was developed[184].” Add space between word and reference.
  • Line 1623: “… was also determined[155].” Add space between word and reference.
  • Line 1636: “… alloys[155].” Add space between word and reference.
  • Line 1652: “… ancient ceramics[174,186].” Add space between word and reference.
  • Line 1686: “… ns-LI-O-TOFMS instrument[187].“Add space between word and reference.
  • Line 1733: “… is considered[190]” Add space between word and reference.
  • Line 1787: “Conflicts of Interest: “The authors declare no conflict of interest.”” Should be “Conflicts of Interest: The authors declare no conflict of interest.”

*Some references are typos or missing page information. Author should fix it.

Author Response

Dear Editor, Dear Reviewer,

In the name of all authors, I thank you for the constructive feedback and the inputs concerning our article, which allowed us to improve the text and presentation. We have added requested changes and further improve the text. The changes are highlighted in yellow in the revised manuscript. We hope that the current version is matching your expectations and look forward for a publication of our article in your journal.

Reviewer #1

‘Tulej et al. reviewed comprehensive information on background, development, and various application of MS using femtosecond lasers. Extensive information is expected to provide useful information for understanding the advantages of using femtosecond lasers in the TOF-MS field, as well as the contents of the prospects to greatly contribute to related fields. I recommend publication of this review paper after revision of the minor parts below.’

  • We thank kindly reviewer for the comment. In following we have make the corrections as suggested.
  • line 90: "Thanks to the efforts of a few groups, the development of LIMS did not stop after a hibernation period in the late 1990s.". I suggest removing "Thanks to the efforts of a few groups" for readability.
  • We have remove requested part from the text
  • lines 139-150: Authors should add some references.
  • We added references to these lines as requested see text
  • line 193:" ... become considerable High thermal energy electrons..." should be " ... become considerable high thermal energy electrons...
  • We have make suggested changes

  • line 231: "[49][38]" reference notation should be corrected.
  • The corrections were made
  • line 235-236: "Figure 1 compares schematically different phases in laser ablation by ns- and fs-lasers." In figure 1 there is no ns- and fs-laser comparison
  • We have removed the sentence and included appropriate one.
  • line 260: "...intensity across the ablation area[44]." Add space between word and reference

corrected

  • line 269: "Early studies by Hergenröder et al. (2006) [19] compared ..." Author should exclude the year notation from the sentence.

corrected

  • line 272: "...gold and copper samples by Amoruso et al. (2002) [75], although..." Author should exclude the year notation from the sentence.

corrected

  • line 273-274: "ratios obtained by Hergenröder et al. (2006) [19]." Author should exclude the year notation from the sentence.

corrected

  • line 290: “(see Fig. 2 )” should be “(Fig. 2)”

corrected

  • line 298: “(see Table 1)” should be “ (Table 1)”

corrected

  • line 303: “(see also Fig.2)” should be “(Fig. 2)”

corrected

  • line 310-311: “by the Huang group (Zhang et al., 2013)” should be excluded.

corrected

  • Line 410-453: Authors should add some references.

corrected

  • Line 463:” … and ionization efficiency[87].” Add space between word and reference

corrected

  • Line 467:” … in material science[99].” Add space between word and reference.

corrected

  • Line 488: “… this spectroscopic method[39].” Add space between word and reference.

corrected

  • Line 493: “…over the plasma parameters[99,100].” Add space between word and reference.

corrected

  • Line 506: “…in recently reported studies[105].”Add space between word and reference.

corrected

  • Line 513: “…intensity tuning) are included[85].”Add space between word and reference.

corrected

  • Line 531: “self-focusing threshold (see also discussion in [44]).” Should be “self-focusing threshold [44]. ”

corrected

  • Line 535: “…(KTiOPO4)[106,107]. In” Add space between word and reference.

corrected

  • Line 541-548: Authors should add some references.

corrected

  • Line 558-574: Authors should add some references.

corrected

  • Line 578-588: Authors should add some references.

corrected

  • Line 621: “… during beam transport (see also discussion in [44]).” should be “… during beam transport [44]. ”

corrected

  • Line 636: “… spectrometric technique (see also discussion in [114]).” should be “… spectrometric technique [114]. ”

corrected

  • Line 756: “(see Fig.4)” should be “(Fig. 4)”

corrected

  • Line 799: “The DP unit[85] consists of two…” Add space between word and reference.$

corrected

  • Line 816: “… centrosymmetric anode rings[129].” Add space between word and reference.

corrected

  • Line 820: “… be measured simultaneously[22].” Add space between word and reference.

corrected

  • Line 832: “… set to improve spectra quality[131].” Add space between word and reference.

corrected

  • Line 960: “… of ppb levels[125].” Add space between word and reference.

corrected

  • Line 1124: “aliquots of the sample (see Table 3) [155].” Should be “aliquots of the sample (Table 3) [155].”

corrected

  • Line 1175: “… abundance[22,125].” Add space between word and reference.

corrected

  • Line 1222: “… lasers[164]”Add space between word and reference.

corrected

  • Line1240: “…56 and 57 nm, respectively, [167],[166]” reference notation should be corrected.

corrected

  • Line 1242: “(see also Fig. 10a)” should be “(Fig. 10a)”

corrected

  • Line 1298: “…double pulse ion source[85].) ”Add space between word and reference.
  • corrected
  • Line 1383; “… meteoritic sample[128].”Add space between word and reference.

corrected

  • Line 1410: “… imaging capability of fs-LMS[36].”

corrected

  • Line 1441 “… microscopic studies[33].”

corrected

  • Line 1452: “(Error! Reference source not found.)” deleted this sentence and add the reference.

Corrected: missing reference to Fig 12.

  • Line 1505: “… ablation spots[180]. Ns-LMS mapping” should be “… ablation spots [180]. ns-LMS mapping” Add space between word and reference.

corrected

  • Line 1512: “… composition[181].” Add space between word and reference.

corrected

  • Line 1595: “… was developed[184].” Add space between word and reference.

corrected

  • Line 1623: “… was also determined[155].” Add space between word and reference.

corrected

  • Line 1636: “… alloys[155].” Add space between word and reference.

corrected

  • Line 1652: “… ancient ceramics[174,186].” Add space between word and reference.

corrected

  • Line 1686: “… ns-LI-O-TOFMS instrument[187].“Add space between word and reference.

corrected

  • Line 1733: “… is considered[190]” Add space between word and reference.

corrected

  • Line 1787: “Conflicts of Interest: “The authors declare no conflict of interest.”” Should be “Conflicts of Interest: The authors declare no conflict of interest.”

corrected

*Some references are typos or missing page information. Author should fix it.

The references are systematically corrected

Reviewer 2 Report

The use of fs pulses to increase ionization is of great value to the field of surface MS. This review further supports the concept that fs pulse can increase ion yield, thereby increasing the spatial resolution of MS and decreasing matrix effects. 

An issue in the publication is the limited explanation of the fundamental processes that facilitate the observed improvements. See page 7, where the authors explain the physics of the process yet use a misattributed citation and misleading wording.

Throughout the publication, the authors make claims that are not supported by references; for example, see page 12, where the authors claim without reference that wavelength has no bearing on the experiment, a concept that is repeated throughout the article. 

While a great variety of topics are covered in this viewer, the work encompasses the authors' primary work. More representation of the wider body of literature is needed. 

Section 3.2 may benefit from comparing traditional ion bombardment to laser ablation. 

The authors mention the goal of the development of an in situ system. This is a noble pursuit that would be very valuable, yet the field of in situ XPS has recently achieved 1-atmosphere measurements after almost 40 years of development. While other traditional UHV techniques may require less than 40 years of development, there are more immediate developmental steps. The publication would also benefit from the authors including perspective on the development of the technique over a shorter time frame. The review would be further improved if it contained the current limitations as well. 

Formating edits are needed; see attached pdf. Please change all sub-figure designations to be constant, i.e. (a) or a). 

Author Response

Dear Editor, Dear Reviewer 2,

In the name of all authors, I thank you for the constructive feedback and the inputs concerning our article, which allowed us to improve our contribution significantly. The changes are highlighted in yellow in the revised manuscript. We hope that the current version is matching your expectations and look forward for a publication of our article in your journal.

  1. ‘The use of fs pulses to increase ionization is of great value to the field of surface MS. This review further supports the concept that fs pulse can increase ion yield, thereby increasing the spatial resolution of MS and decreasing matrix effects.’
  • Thank you for the comment. The statement made here is in our opinion a bit too far simplification. If we may specify, the application of fs-laser for the ablation improves control over the ion production, ion yield and ion stoichiometry. With application of short pulse one has also an improved understanding over the ablation mechanism and how to control ion yields and cluster production (e.g., double pulse option with a certain delay between the pulses). Fs-radiation stability allows much better control over ablation rate and control over the depth profiling resolution. Depth profiling analysis further can be used to conduct analysis of sample layering, grains and as shown recently, the correlation between the isotopes or atoms can be achieved leading to improved quantification. Reduced heat effects induced during surface ablation contribute to such a study as well. The ion fluctuations are considerable reduced as well so that relative atomic concentrations can be correlated at every laser shot. Lateral resolution can be improved further due to the reduction of the heat affected zone. fs-LIMS is less dependent on the physical and chemical properties of the material (compared to the case where ns-laser ablation is applied). This is due to nonlinear absorption or multiphoton absorption allowing for efficient absorption even of IR photons by the materials typically absorbing in UV range (multiphoton absorption). Dependence on the absorption wavelength still is expected but the magnitude of this dependence is lower compared to ns-laser ablation because of aforementioned multiphoton absorption efficiency. Also, other dependencies on laser ablation source parameters such as pulse duration, laser wavelength, laser pulse energy, laser irradiance are expected and they are observed but they are less critical compared to that for longer pulse lasers.
  1. ‘An issue in the publication is the limited explanation of the fundamental processes that facilitate the observed improvements.
  • The most of reviewed papers here emphasize difference between the ablation by ns- and fs-lasers, justifying observed phenomena and referring to relevant fundamental processes. We have also reference previous review papers on the LIMS technique to not repeat this information. Furthermore, there are also well-written mongraphs e.g., Gamaly providing necessary background expanding on theoretical principles underlying fs-laser ablation. A substantial number of diagnostic experimental studies and studies of the specific ablation mechanism can be in fact reviewed independently including the theoretical modelling for more detailed understanding of the processes (Gamaly monograph does this job well). It will difficult here to expand studies of these group in one section considering the contributions from all contributing groups. We added references for the reader to some of these investigations. We would be happy to expand on this subject upon the reviewer’s specific request.
  1. See page 7, where the authors explain the physics of the process yet use a misattributed citation and misleading wording.
  • Thank you for this correction. We improved this part by adding typically important avalanche mechanism and becoming important in femtosecond radiation multiphoton absorption or nonlinear absorption. For the reader’s convenience, we refer to the book section by F. Hagelund introducing the principles underlying solid state physics including metals, semiconductors and insulators in the frame of band theory relevant in laser matter interactions. Theoretical background relevant to femtosecond physics of laser-matter interaction can be found in the Grossmann monograph. More detailed theoretical analysis of the fs-laser interaction vs. ns-laser interaction can be found in the Gamaly monograph. Some misleading wording was corrected in the text.
  • Added section: Two major processes responsible for absorption in metals and dielectrics involve the intra-band and inter-band transitions [43, 44, 55, 56]. The first include the electrons excitation and heating in metals whereas the second is important for absorption in dielectrics. And involves single/multi-photon absorption and avalanche acceleration of electrons in valence band to the energy exceeding band gap. Dielectric subsequently can considered to be in the metal-like state [43]. Usually a few seed electrons are present in the valence band. Although the direct photon absorption is typically small, they gain energy by accelerations and collisions in laser field. Once they gain energy in excess of the band gap, can further collide with the electrons in the valence gap to transfer them for excitation into conduction band. Thus, the mechanism involves an avalanche of ionisation events. The other important ionization process that contributes into the total ionization rate is multi-photon ionization. This process is readily enhanced while fs-laser radiation is applied. Probability of simultaneous absorption of several photons (multi-photon absorption) is increasingly higher in case of femtosecond radiation than in case of  ns-laser radiation [43,44,55]. Through multiphoton absorption induced by fs-lasers, a sufficiently large amount of energy can be deposited to cause ablation, even in materials that are transparent to the applied laser wavelength [57]. Typically, the light used needs to be of some sufficiently high enough energy (depending on the optical properties of the material) to be absorbed by the ablated material. Then the power density needs to be high enough to ionise all ablated material. For most materials ablated with ∼100 fs laser pulses, laser intensity  in the range from 1012 to 1013 W/cm2 is applied [44]. At these conditions multiphoton absorption becomes important. The ionisation time duration (separation of electrons from the atoms) is becoming also shorter than the laser pulse duration. Ionization of the sample material already occurs early in the time-span of the radiation interaction (tens offs) [58] and can create a high-density plasma within the nm-thin surface. The primary process of electron excitation by the ultra-short laser pulse occurs in complete non-equilibrium conditions.
  •  
  1. Line 181: This requires a citation as it is the reason why fs pulse assist in the measurement. ‘

We would like to emphasise that the fs-laser ablation mechanism involves nonlinear absorption. This improves ablation efficiency and reduces dependence on the physical (optical) properties of the material. See also answer 3.

Line 185 ref [54]: reference 54 uses 16 ns pulse light and does not mention multi-photon absorption’ 

  • Thank you for this correction. The wrong citation was inserted. The citation of the review paper by Bally and Schou is added now instead.

‘Line 186: This is stated poorly; the light used needs to be of some high enough energy to be absorbed by the ablated material. Then the power density needs to be high enough to ionize all ablated material. ‘

  • We agree with the reviewer in general. With the reviewer’s help we have corrected the sentence in question. Nevertheless, at these power densities the absorption can occur efficiently via combining a few photons so the energy required for the absorption is sufficient without using one more energetic photon although this will be more efficient mechanism of excitation/ionisation. For a longer laser pulse duration (ns-laser) the multi photon mechanism is less efficient. For the theoretical background, we refer to e.g., Grossman monograph.
  1. Throughout the publication, the authors make claims that are not supported by references; for example, see page 12, where the authors claim without reference that wavelength has no bearing on the experiment, a concept that is repeated throughout the article. 
  • Thank you for this comment. On the page 12 we stated that there are no systematic fs-LIMS studies of ion yield as a function of wavelength but it will be important to conduct such ones. It was never our intention to convince reader that there is no dependence on wavelength. What we are saying that the dependence on wavelength is weaker compared to ablation by ns-laser sources and this is accounted to improved absorption of laser radiation by nonlinear laser matter interaction mechanism. We have conducted also further studies with 3 wavelengths (now prepared for publication) showing in fact that there is improvement of the ablation efficiency and ion yield when the wavelength of fs-laser is shorter.

‘Line 308: As stated above, there are no studies on wavelength effects at fs pulses. Stating there is no effect with photon energy is false; this has not been reported yet. Furthermore, if the effect is observed with contentious light and ns pulse light, it would follow that there should be an effect with fs pulse light. ‘

  • We do not say this in our manuscript so only that this dependence is weaker. Here is Line 308: ‘Due to the nonlinear absorption of fs-radiation, the dependence on material and wavelength is smaller than that of ns-lasers and further improves the UV fs-laser ablation.’

The statement is corrected for a better readability: Due to the nonlinear absorption of fs-radiation, the dependence of ablation efficiency and ion generation efficiency on material and wavelength is observed to be smaller than that of ns-lasers but still exists. While conducting the UV fs-laser ablation again less dependence on material properties and smaller fractionation effects compared to the IR fs-laser ablation can be observed[82,83].

Typically, by means of classical linear absorption process (low intensity radiation sources), the energy of the photon to be absorbed has to match the absorption band. With fs-laser radiation even low energy IR photons can be used and will be absorbed by the material absorbing in UV range. This occurs by multiphoton absorption which efficiency improves significantly when focused fs-laser radiation is applied. The probability of n-photon absorption is proportional to In, with I being the radiation intensity.

We are preparing this time manuscripts with the experimental results obtained at 3 wavelength 258, 378 and 775 nm (150 fs laser pulse), and also double pulse ion source. These results show dependance on the wavelength. Generally, an improved ablation and reduced fractionations for a shorter wavelength.

  1. While a great variety of topics are covered in this viewer, the work encompasses the authors' primary work. More representation of the wider body of literature is needed. 
  • Thank you for the comment. To the best of our knowledge we discuss the results of all groups conducting fs-LIMS application studies. In fact, we were productive in recent years but Huang group was as well. We would gladly add the contributions from the other groups, if the reviewer would be more specific which publications we did not consider.
  1. Section 3.2 may benefit from comparing traditional ion bombardment to laser ablation. 
  • We would be happy to include the ion bombardment but we are not sure if the reviewer meant really section 3.2. Here we discuss TOFMS used in fs-LIMS systems.
  1. The authors mention the goal of the development of an in situ system. This is a noble pursuit that would be very valuable, yet the field of in situ XPS has recently achieved 1-atmosphere measurements after almost 40 years of development. While other traditional UHV techniques may require less than 40 years of development, there are more immediate developmental steps. The publication would also benefit from the authors including perspective on the development of the technique over a shorter time frame. The review would be further improved if it contained the current limitations as well. 
  • We agree with the reviewer comment that the process of developing a portable instrument is long. Our miniature LIMS system was introduced in 2003 (Rohner et al., 2003) and this prototype was ready for the mission to Mercury. Our group is developing LIMS for space research for in situ measurements on planetary surfaces. The pressure at the lunar surface is in the order of 10-10 The miniature LIMS, the LAZMA instrument was accepted on these missions and the flight instrument is currently ready. This instrument will be flown on LUNA missions soon. The vacuum of 10-7 mbar can be obtained using specially developed miniature turbo-molecular pump. For the martian environment a turbomolecular pump was developed and applied already on the NASA rover mission. We have systematically updated in previous publications the status of the LIMS to be a portable instrument: Rohner et al., 2003, Tulej et al., 2013 and in a number of more recent publications.
  1. The review would be further improved if it contained the current limitations as well. 
  • We have mentioned several limitations while reading through the instrumental part and in the applications part. Also, we notice also a need for the future developments which are needed for the improvement of the quantitative performance of this technique. Just to recall a few such limitations. One of the serious problem is a surface and space charge effects can influence the performance of LIMS systems introducing temporal jittering of the waveform. Currently, it is not easy to measure and save single spectra; one rather summed up a few spectra first and then save this accumulated spectrum (other limitation). By this procedure even small shifts introduced by jittering can lead to the mass peak broadening limits the mass resolution. Important limitation can be inefficient ion transfer from the ion source to mass separation unit (TOF) and an ion detector. In linear TOF MS, 100% ion transmission can be achieved by the optimised ion optics. Nevertheless, applying rectangular ion extraction from the source, ion transmission through the collisional cell and rectangular ion extraction into TOF tube allows only a fraction of ions to be analysed (can reduce sensitivity, lead to fractionation effects). On the other hand, a control over the ion current that the ion detector not becoming saturated is other problem. We have emphasized also that currently applied Gaussian pulse width leads to several limitation regarding ablation process. The threshold energy is not well defined, The shape of crater is far from cylindrical and can cause severe problems in the analysis of depth profiling. We have discussed the pulse shaping methods which can improve this situation. An access to short pulse deep UV or X-ray spectral region further can improve the ablation etc. Other problem is also clearly stated that the mass resolution of current mass spectrometers is limited and to resolve isobaric interferences one would need to increase mass resolution further. Also mass spectrometric data analysis can be improved with new concepts and statistical approaches. 
  1. Formatting edits are needed; see attached pdf. Please change all sub-figure designations to be constant, i.e. (a) or a). 
  • Thank you for the comment. We have made the required corrections to each figure.

With my best regards,

PD Dr. Marek Tulej

Round 2

Reviewer 2 Report

The authors have addressed all of the previous comments, most of which are incorporated into the manuscript. While including all the comments in the paper would be appreciated, the review in its current form provides a comprehensive overview of the current state of the field.